# PAM-flexible Engineered FnCas9 variants for robust and ultra-precise genome editing and diagnostics

Sundaram Acharya [1,2] ✉, Asgar Hussain Ansari [1,2], Prosad Kumar Das [1],
Seiichi Hirano[3], Meghali Aich[1,2], Riya Rauthan[1,2], Sudipta Mahato[4,5],
Savitri Maddileti[4], Sajal Sarkar[1,2], Manoj Kumar[1,2], Rhythm Phutela[1,2],
Sneha Gulati[1], Abdul Rahman[1], Arushi Goel[1,2], C. Afzal[1], Deepanjan Paul[1],
Trupti Agrawal [4,5], Vinay Kumar Pulimamidi[4,6], Subhadra Jalali[7],
Hiroshi Nishimasu [8,9,10], Indumathi Mariappan [4], Osamu Nureki [3],
Souvik Maiti[1,2] & Debojyoti Chakraborty [1,2] ✉

The clinical success of CRISPR therapies hinges on the safety and efficacy of Cas proteins. The Cas9 from *Francisella novicida* (FnCas9) is highly precise, with a negligible affinity for mismatched substrates, but its low cellular targeting efficiency limits therapeutic use. Here, we rationally engineer the protein to develop enhanced FnCas9 (enFnCas9) variants and broaden their accessibility across human genomic sites by ~3.5-fold. The enFnCas9 proteins with single mismatch specificity expanded the target range of FnCas9-based CRISPR diagnostics to detect the pathogenic DNA signatures. They outperform *Streptococcus pyogenes* Cas9 (SpCas9) and its engineered derivatives in on-target editing efficiency, knock-in rates, and off-target specificity. enFnCas9 can be combined with extended gRNAs for robust base editing at sites which are inaccessible to PAM-constrained canonical base editors. Finally, we demonstrate an *RPE65* mutation correction in a Leber congenital amaurosis 2 (LCA2) patient-specific iPSC line using enFnCas9 adenine base editor, highlighting its therapeutic utility.

Like *Streptococcus pyogenes* Cas9 (SpCas9), *Francisella novicida* Cas9 (FnCas9) recognizes the minimal 5′-NGG-3′ protospacer adjacent motif (PAM) yet shows a much higher sgRNA sequence-dependent specificity when interrogated with DNA substrates[1-4]. Although high-fidelity versions of SpCas9 have been designed and validated in multiple systems, their editing efficiencies have generally dropped significantly as compared to the wild-type enzyme[5-7]. To circumvent these issues, in recent years, alternate high-efficiency Cas systems from other microorganisms have been utilised for genome editing[8-12]. Notably, none show consistent editing efficiencies higher than SpCas9[5-7], and the

[1]CSIR-Institute of Genomics & Integrative Biology, Mathura Road, New Delhi 110025, India. [2]Academy of Scientific & Innovative Research (AcSIR), Ghaziabad 201002, India. [3]Department of Biological Sciences, Graduate School of Science, The University of Tokyo, 7-3-1 Hongo, Bunkyo-ku, Tokyo 113-0033, Japan. [4]Center for Ocular Regeneration, Prof. Brien Holden Eye Research Centre, Hyderabad Eye Research Foundation, LV Prasad Eye Institute, Hyderabad 500034 Telangana, India. [5]Manipal Academy of Higher Education, Manipal, Karnataka, India. [6]Schepens Eye Research Institute, Massachusetts Eye and Ear, Harvard Medical School, Boston, MA 02114, USA. [7]Srimati Kannuri Santhamma Centre for vitreoretinal diseases, Anant Bajaj Retina Institute, Kallam Anji Reddy Campus, L V Prasad Eye Institute, Hyderabad, Telangana, India. [8]Department of Chemistry and Biotechnology, Graduate School of Engineering, The University of Tokyo, 7-3-1 Hongo, Bunkyo-ku, Tokyo 113-8656, Japan. [9]Research Center for Advanced Science and Technology, The University of Tokyo, 4-6-1 Komaba, Meguro-ku, Tokyo 153-8904, Japan. [10]Inamori Research Institute for Science, 620 Suiginya-cho, Shimogyo-ku, Kyoto 600-8411, Japan.
✉e-mail: acharyasundaram.ac@gmail.com; debojyoti.chakraborty@igib.in

majority of these enzymes have a PAM requirement that is more complex and less available in the human genome than SpCas9 (Supplementary Data 1). This limits the number of possible sites accessible for therapeutic correction[13–18]).

In earlier studies, we and others reported that FnCas9 has a very high intrinsic specificity, resulting in dissociation from off-targets presented in vitro[4,19]. In contrast, SpCas9 and its high-fidelity variants remain bound to off-target sites in a cleavage incompetent fashion, a property that might cause non-specific off-targeting outcomes from such regions[20–22]. Where nuclease activity is not utilised, such as in the realm of base editing, constraints introduced by the targeting window of the base editors, concerning the nearest PAM, have necessitated engineering Cas9 proteins with altered PAM specificities to access nucleobase targets on a case-to-case basis[23]. Taken together, there is an unmet need for Cas effectors that can combine high activity and specificity on one hand, and flexibility in base editing on the other, especially when a favorable PAM is not present in the vicinity of the editing window.

In this study, we have rationally engineered FnCas9 by modifying its WED-PI domain and phosphate-lock loop (PLL) to develop enhanced (en) FnCas9 variants. We have identified and characterized three kinetically enhanced, PAM-flexible enFnCas9 variants (en1, en15 and en31) without altering the intrinsic DNA interrogation specificity of FnCas9, as reported earlier by our group[4]. enFnCas9 variants expand the range of single-nucleotide variant (SNV) detection by FnCas9-based CRISPR diagnostics (CRISPRDx) platforms. PAM-flexible robust cellular editing, improved HDR-mediated knock-in rate, and single nucleobase specificity underscores the superior performance of enFnCas9 variants in human cells. Furthermore, the compatibility of enFnCas9 variant, en31 with extended (x-) or super-extended (sx-)-gRNAs, enables tuning of adenine base editing window for adenine base editors (ABEs) by en31-ABEmax8.17 without the requirement of PAM engineering to target intended bases. This attribute expands the range of pathogenic single-nucleotide polymorphisms (SNPs) targeting by en31-ABE. Finally, we demonstrate the precise correction of a Leber congenital amaurosis, type 2 (LCA2) disease-associated point mutation in the *RPE65* gene which results in the generation of a premature stop codon in the disease condition. We show restoration of the full-length protein expression in patient-specific iPSC-derived retinal pigmented epithelium using the en31-ABEmax8.17. Overall, we show engineered enFnCas9 variants with broad applications in therapeutics and diagnostics.

## Results

### Engineering and characterization of FnCas9 variants for enhanced activity

To investigate if FnCas9's high DNA binding specificity is reflected on a genome-wide level, we compared the genomic interrogation status of catalytically inactive (dead, d) dSpCas9 and dFnCas9 proteins after ensuring comparable protein expression in human cells (Supplementary Fig. 1b, 2a). We targeted the *c-Myc* locus where similar cellular editing efficiencies between SpCas9 and FnCas9 were observed previously[4]. Using chromatin immunoprecipitation followed by massively parallel sequencing (ChIP-seq)[24–26], we found that although both dSpCas9 and dFnCas9 were tightly bound to the on-target sites, dSpCas9 showed promiscuous binding at multiple off-targets (27 sites, 0.01 FDR) across the genome, even at sites with up to 6 mismatches in the sgRNA (Supplementary Fig. 2b–d, Supplementary Data 2). Interestingly, all the 27 dSpCas9 off-target sites showed greater enrichment than the on-target. In contrast, dFnCas9 was bound to 6 off-target sites (0.01 FDR) and all of these showed at least 1.2-fold lower enrichment than the on-target (Supplementary Fig. 2b–d, Supplementary Data 2). This high specificity of binding thus presented an attractive scenario for structure-guided engineering, using FnCas9 protein as the chassis to develop a robust, programmable CRISPR system without compromising its intrinsic target interrogation specificity for genome editing applications.

FnCas9 is evolutionarily divergent to SpCas9 and harbors structural dissimilarities such as unique interactions between the RuvC and REC3 domains, and the PI and WED domains with the latter sharing contacts with the REC1 and REC2 domains[1,3,4]. However, PAM recognition is conserved among Cas9 orthologs, which trigger directional target DNA unwinding, R-loop formation and expansion, which eventually lead to reorientation of the HNH endonuclease domain to DNA cutting and concomitant RuvC activation leading to concerted DNA cleavage[27]. Recent mechanistic studies showed that the directional PAM-duplex DNA unwinding serves as the rate-limiting checkpoint of Cas9 action and a conformational switch discriminates Cas9 DNA binding and cleavage events[21,28–32]. Moreover, the loss of nucleobase-specific interaction between the target DNA and Cas9 was reported to be rescued by base non-specific Cas9 interactions[3,33]. Thus, we reasoned that stabilizing FnCas9:DNA duplex binding by introducing base non-specific interactions between PAM duplex and the protein might improve FnCas9 nuclease activity without altering its intrinsic specificity. Additionally, to investigate the optimal spacer length in DNA cleavage activity of FnCas9, we performed an in vitro cleavage assay using a previously reported target DNA harboring a stretch of guanines with FnCas9 RNP containing gRNAs of variable lengths ranging from 20 to 24 nucleotides (g20-g24, hereafter referred as extended-gRNA, x-gRNA)[34]. Interestingly, we observed the lowest activity with the canonical g20, while x-gRNAs exhibited an enhanced DNA cleavage rate, with g21 inducing the fastest rate of cleavage (Supplementary Fig. 3a). We have used g21 in all subsequent assays unless otherwise stated.

Next, we engineered 49 different FnCas9 variants, guided by its crystal structure bearing mostly single amino acid substitutions in the WED-PI domain to introduce distinct PAM duplex DNA contacts (Fig. 1a, Table 1). Subsequently, we measured in vitro DNA cleavage activities of the FnCas9 variants against a DNA target containing 5'-GGG-3' PAM (where FnCas9 was shown to be least active)[3] and performed DNA cleavage experiments with the engineered variants (Supplementary Fig. 3c). Interestingly, we found some of the variants showing higher rate of DNA cleavage compared to FnCas9 (Supplementary Fig. 3c). Recent reports have suggested that high-fidelity SpCas9 variants have slower enzyme kinetics and lower editing efficiencies[5–7]. Given that FnCas9 is an enzyme with slow cleavage rate and high fidelity[4], we focused on enhanced (en) FnCas9 variants that exhibit a faster cleavage rate with minimal structural alterations to ensure its intrinsic specificity remains unchanged.

A subset of nine enFnCas9 variants (containing single/combinatorial mutations) were selected for downstream experiments satisfying these criteria (Fig. 1b, Supplementary Fig. 4a). Among them, three variants (en1, en15 and en31) showed at least a 2-fold higher cleavage rate than the wild-type (WT) FnCas9 (Fig. 1b). Structural models revealed that en1 (E1369R) and en15 (E1603H) variants create additional interactions with the backbone phosphate group of dA(−1) in the target DNA strand and stabilize dT(1) in the non-target strand, respectively (Supplementary Fig. 3d). In the triple mutant (G1243T/E1369R/E1449H, en31), the G1243T makes hydrogen bonding with +1 phosphate of the phosphate-lock loop (PLL) in the target strand, initiating DNA unwinding[2]. Additionally, E1369R/E1449H engage in electrostatic interactions with the phosphate backbone between dC(−2) and dA(−1) in the target DNA strand[3] (Supplementary Fig. 3e). Notably, the G1243T (en4) variant alone is insufficient to significantly enhance the rate of DNA cleavage compared to the wild-type FnCas9 (Fig. 1b; Supplementary Fig. 3c).

Intrigued by the improved kinetic activity of the variants, we tested the cleavage efficiency of two of the enFnCas9 variants (en15 and en31) using super-extended (sx)- gRNAs with spacer lengths ranging from 26 to 28 nucleotides (g26-g28, hereafter referred to as sx-gRNA). We confirmed similar cleavage efficiencies as g21, suggesting the compatibility of enFnCas9 variants with sx-gRNAs (Supplementary Fig. 3b). This could be attributed to the elongated REC3 domain, which

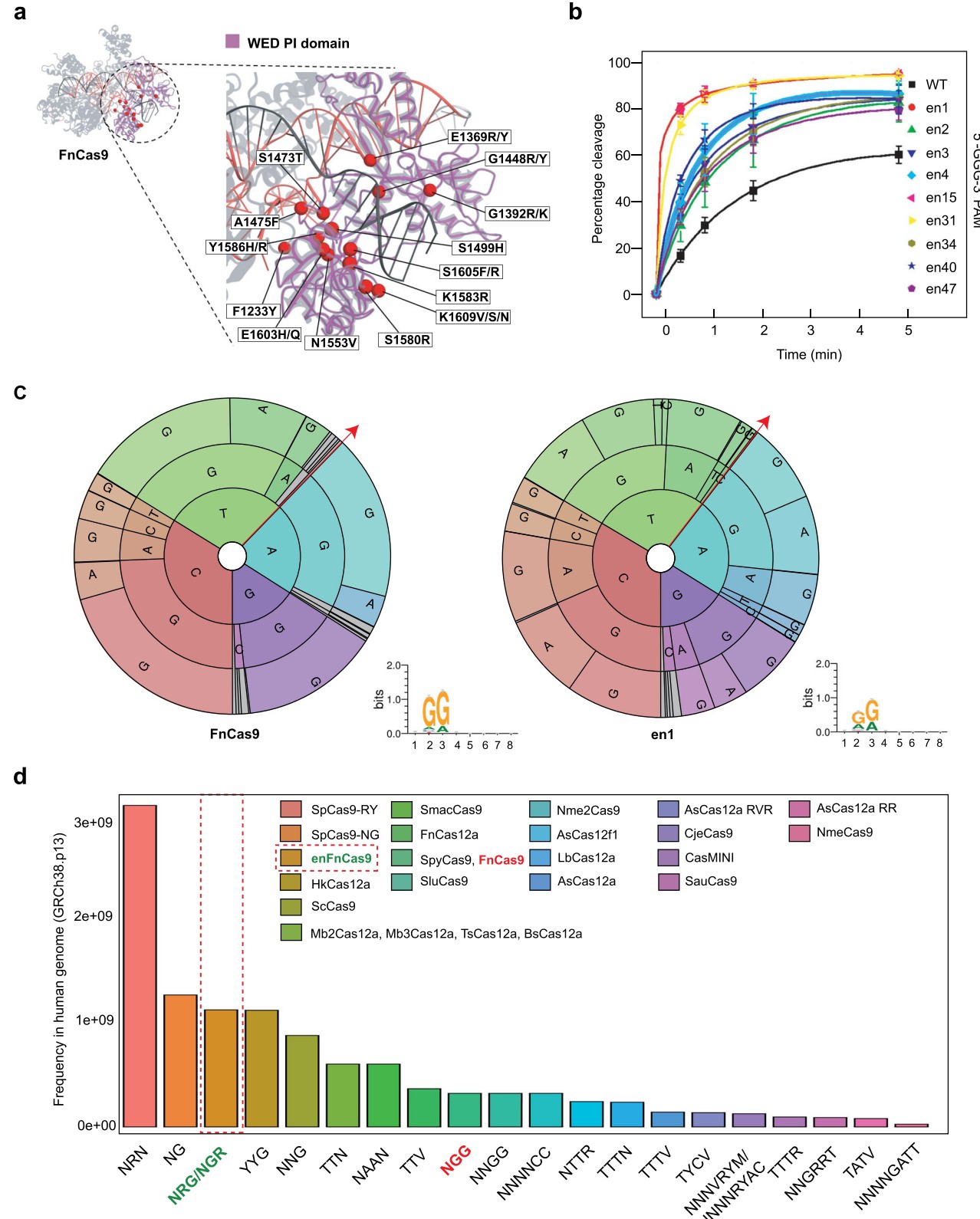

interacts with the nucleobases away from the PAM, as visualized in its crystal structure[3]. To our knowledge, similar observations have not been reported so far for other Cas9 proteins until recently, during the revision of this work[35]. We speculated that this feature might offer further enhancement of specificity and nucleobase accessibility away from PAM as shown later.

## Engineered FnCas9 variants relaxed the PAM requirements, leading to altered PAM specificity and enhanced activity at non-canonical PAM

The crystal structure of FnCas9 revealed that the PAM duplex is nestled in the FnCas9 WED-PI domain. In the 5′-NGG-3′ PAM sequence, the major groove dG(2) and dG(3) on the non-target strand are recognized

**Fig. 1 | Engineering and characterization of FnCas9 variants for enhanced activity and its alterd PAM specificity. a** FnCas9 crystal structure in complex with sgRNA-DNA (PDB: 5B2O) in ribbon model with highlighted WED-PI domain marked in dotted circle. Zoomed inset shows amino acid residues substituted for protein engineering. Some residues are not shown due to clarity. Full details are available in Table 1. **b** In vitro cleavage assay of WT (FnCas9) and a subset of nine enFnCas9 variants on a 5′-GGG-3′ PAM containing PCR linearized DNA substrate, expressed as percentage cleavage (y-axis) as a function of time (x-axis). Error bars represent mean ± SD of $n = 3$ independent experiments. **c** The PAM wheels and sequence logos showing the results obtained from PAM discovery assay for FnCas9 and en1.

Individual sections of the pie charts in the PAM wheels with ≤ 2% depletion enrichment are shown in gray. Based on the inner to the outer circle in the PAM wheels map, the PAM reads away from the target region in the 5′ to 3′ direction, as shown by red arrows. **d** Bar plot showing the availability of PAMs of respective Cas effectors in the human genome expressed as the frequency in the human genome on the y-axis and PAM sequence on the x-axis. Respective 5′-NGG-3′ and 5′-NRG/NGR-3′ PAMs of FnCas9 and enFnCas9 are highlighted in red and green accordingly. Red dotted box highlights the PAM preferences for a subset of enFnCas9 variants. The Source Data are provided in the source data file.

by R1585 and R1556 respectively, through bidentate hydrogen bonds, while dN(1) remains free from base-specific protein contacts[3]. Towards developing a 5′-NG-3′ PAM-specific FnCas9 variant, earlier work reported that substituting R1556A abolishes the protein function. Partial rescue of functional activity was achieved by incorporating base non-specific interactions with E1369R and E1449H mutations, resulting in the creation of the RHA FnCas9 variant capable of targeting 5′-YG-3′ PAM[3]. However, we observed a very low rate of DNA cleavage in vitro when compared with FnCas9 using a substrate containing a 5′-TGG-3′ PAM, consistent with earlier observations[3] (Supplementary Fig. 4b).

Given the major groove adenine:glutamine contact and the preference of threonine for thymine base[2,36], we substituted R1556Q (en49) and R1556T (en17) to create variants with 5′-NGA-3′ and 5′-NGT-3′ PAM specificity, respectively. However, we did not observe enzymatic activity in vitro (Supplementary Fig. 3c), consistent with earlier observations made with R1556A[3]. This confirms that the interaction between R1556 with dG(3) of non-target strand of the PAM duplex is indispensable for FnCas9 functional activity. It also necessitates exploring alternative ways to modify PAM specificity while maintaining robust enzymatic activity.

Towards developing PAM-relaxed enFnCas9 variants, we found that earlier reports have shown engineered SpCas9 variants often establish additional phosphate backbone interactions, enabling these proteins to recognize non-canonical PAMs[37,38]. We reasoned that a similar phenomenon may occur for enFnCas9 variants when creating additional interaction with the PAM duplex. To this end, we tested a subset of enFnCas9 variants with improved activity, as identified in the initial screening with canonical PAM, to investigate the potential alterations in PAM specificity using a substrate containing 5′-GGA-3′ PAM. It is noteworthy that the moderate activity of FnCas9 on 5′-NGA-3′ PAM has been attributed to R1556 which recognizes dA(3) through a single hydrogen bond[3].

Encouragingly, we discovered five enFnCas9 variants (en1, en15, en31, en34 and en40) showing enhanced activity on DNA substrate containing 5′-GGA-3′ PAM when compared with FnCas9 (WT) (Supplementary Fig. 4c, d). Among them, en31 and en34 exhibited the highest cleavage rates (Supplementary Fig. 4d). To comprehensively determine the landscape of PAM specificity, we performed an in vitro PAM discovery assay using a plasmid library containing a randomized 8 bp sequence ($4^8 = 65,536$ combinations in total) in the PAM region (Supplementary Fig. 5a). Deep sequencing of the PAM-depleted library revealed that all tested enFnCas9 variants showed more flexible recognition at the second and third nucleotide positions compared to FnCas9, resulting in PAM recognition of 5′-NRG/NGR-3′ (Fig. 1c, Supplementary Fig. 5b–e). Importantly, this PAM alteration expanded the accessibility of enFnCas9 variants across the human genome ~3.5-fold over the wild-type FnCas9 (Supplementary Fig. 4e), placing enFnCas9 just below SpCas9-RY[39] and SpCas9-NG[33] (Fig. 1d, Supplementary Data 1).

**enFnCas9 variants expand the target range and sensitivity of SNV detection**

The remarkable intrinsic specificity of FnCas9 to single nucleotide mismatches in the target has applications both in disease diagnostics

and disease correction[4,40,41]. At the level of diagnostics, FnCas9 has recently been utilised for paper strip-based robust diagnostics of nucleic acid targets through the FnCas9 Editor Linked Uniform Detection Assay (FELUDA)[40] and Rapid Variant Assay (RAY) platforms[41]. In contrast to collateral cleavage-based platforms employed by Type V effectors (such as Cas12a[42] or Cas12f[43]) or Type VI effectors (such as Cas13[44]), FELUDA and RAY uses the specificity of direct FnCas9:DNA binding as a lateral-flow readout through a combination of FAM-labeled FnCas9:sgRNA complex and paper-strip chemistry[40,41] (Fig. 2a). Importantly, FnCas9 showed comparable resolution of single-nucleotide variant (SNV) diagnosis (4.4-fold) as AaCas12b (4.6-fold) and Cas14a1 (5.1-fold) both of which belong to type V DNA targeting Cas systems and have been reported to have higher intrinsic specificity than SpCas9[10,11,15,16,45] (Supplementary Fig. 6a). Taken together, this underscores the utility of these variants as a diagnostic platform.

We found that in comparison to FnCas9, enFnCas9 (with 5′-NRG/NGR-3′ PAM specificity)-based CRISPR diagnostics can now cover ~2-fold higher number of reported Mendelian SNVs across the human genome, thereby increasing the scope of detection to more disease-causing variants (Fig. 2b). Expectedly, on a lateral flow strip, all enFnCas9 variants tested (complexed with 20-nt gRNA, g20) showed robust activity on a substrate carrying the non-canonical 5′-NGA-3′ PAM, whereas FnCas9 did not show any signal (Supplementary Fig. 6b).

Since, enFnCas9 variants were constructed by altering residues that stabilize the PAM duplex binding keeping the DNA interacting domains responsible for PAM distal mismatch sensitivity untouched, we speculated that variants should still retain the high single-mismatch specificity as FnCas9. For all the enzymes, tolerance to mismatches was lowest at the most PAM-proximal (1st) and distal (15th to 19th) bases (Supplementary Fig. 6c). However, unlike FnCas9, all the three variants showed increase in the cleavage activity from 2nd to 11th base mismatches of the sgRNA (Supplementary Fig. 6c). This can be attributed to faster cleavage rates of enFnCas9 variants since, even for FnCas9, longer incubation times can lead to substrate cleavage with mismatches in these positions[4]. To determine the impact of these changes on the diagnostic potential of enFnCas9 variants, we selected en31 which showed the broadest activity at altered PAM sites. We investigated if en31 was able to distinguish single mismatches in two targets with pathogenic mutations related to Sickle Cell Anemia and the SARS-CoV-2 Alpha VOC signature (N501Y). Remarkably, en31 accurately distinguished the SNP associated with Sickle Cell Anemia on a lateral flow device with an improved signal discrimination (up to 3.9-fold) as compared to FnCas9 (Fig. 2c, d). We confirmed that the same specificity of SNV discrimination can be extended to both 5′-TGA-3′ and 5′-TGG-3′ PAM-containing substrates associated with SARS-CoV-2 Alpha VOC signature (N501Y) (Fig. 2e, Supplementary Fig. 6d). To test whether the improved signal discrimination is an outcome of stabilization of the PAM-duplex binding by variants, we used catalytically inactive versions of two of the variants (en1 and en15) and performed micro-scale thermophoresis (MST) to determine their DNA binding affinities on a previously reported substrate (VEGFA3) with a 20-nt gRNA[4]. We found that these variants showed stronger DNA binding (Kd = 91.33 ± 29.8 nM for en1, Kd = 49.16 ± 10.96 nM for en15) as compared to FnCas9 (Kd = 170 ± 31.53 nM), with en15 showing ~3.5-fold higher DNA

**Table 1 | Details of FnCas9 variants designed by structure-guided protein engineering**

| FnCas9 variants | Amino acid position | Amino acid change |
|---|---|---|
| en1 | 1369 | E > R |
| en2 | 1449 | E > H |
| en3 | 1369, 1449 | E > R, E > H |
| en4 | 1243 | G > T |
| en5 | 1369 | E > Y |
| en6 | 1392 | G > R |
| en7 | 1392 | G > K |
| en8 | 1448 | N > R |
| en9 | 1448 | N > Y |
| en10 | 1451_1452 | ins V |
| en11 | 1473 | S > T |
| en12 | 1553 | N > V |
| en13 | 1586 | Y > H |
| en14 | 1586 | Y > R |
| en15 | 1603 | E > H |
| en17 | 1369, 1449, 1556 | E > R, E > H, R > T |
| en18 | 1233 | F > Y |
| en19 | 1475 | A > F |
| en20 | 1499 | S > H |
| en21 | 1580 | S > R |
| en22 | 1583 | K > R |
| en23 | 1609 | K > V |
| en24 | 1609 | K > S |
| en25 | 1609 | K > N |
| en26 | 1605 | S > F |
| en27 | 1605 | S > R |
| en28 | 1386_1387 | ins RR |
| en29 | 1586, 1603 | Y > H, E > H |
| en30 | 1392, 1448 | G > K, N > Y |
| en31 | 1243, 1369, 1449 | G > T, E > R, E > H |
| en32 | 1369, 1392, 1449 | E > R, G > K, E > H |
| en33 | 1369, 1448, 1449 | E > R, N > Y, E > H |
| en34 | 1369, 1603 | E > R, E > H |
| en35 | 1369, 1392 | E > R, G > K |
| en36 | 1369, 1448 | E > R, N > R |
| en37 | 1369, 1448 | E > R, N > Y |
| en38 | 1369, 1475 | E > R, A > F |
| en39 | 1369, 1580 | E > R, S > R |
| en40 | 1243, 1369 | G > T, E > R |
| en41 | 1369, 1556 | E > R, R > Q |
| en42 | 1392, 1603 | G > K, E > H |
| en43 | 1448, 1603 | N > R, E > H |
| en44 | 1448, 1603 | N > Y, E > H |
| en45 | 1475, 1603 | A > F, E > H |
| en46 | 1580, 1603 | S > R, E > H |
| en47 | 1243, 1603 | G > T, E > H |
| en48 | 1556, 1603 | R > Q, E > H |
| en49 | 1556 | R > Q |
| en50 | 1369, 1449, 1556 | E > R, E > H, R > Q |

binding affinity (Fig. 2f, Supplementary Fig. 6e). Interestingly, in our previous study, we showed weaker binding of FnCas9 to the same substrate compared to SpCas9 (3.02-fold) under similar conditions[4]. Thus, engineering improved enFnCas9:DNA binding affinity, reaching similar levels as SpCas9 but with superior specificity.

Taken together, enFnCas9 variants have a very high specificity of mismatch discrimination similar to Cas12a or Cas12f/Cas14a but due to their wider PAM accessibility, these can potentially target more genomic sites and pathogenic SNVs for detection.

## Superior cellular genome editing by enFnCas9 variants

The safety of therapeutic genome editing is determined by off-target interrogation of CRISPR effectors. Although Cas12a and Cas12f have higher specificity than SpCas9, their therapeutic success relies on minimum ssDNA cleavage inside the cell such as those formed during replication, homology-directed repair, or transcription[42,46]. Interestingly, Cas12a has been reported to nick off-target DNA substrates with up to four mismatches depending upon the crRNA sequences employed[47]. On the contrary, enFnCas9 does not produce trans-cleavage products, and its high specificity, both at the level of DNA interrogation and cleavage, might be beneficial for safe nuclease-mediated genome editing. Although construction of high-fidelity SpCas9 proteins have improved its overall specificity, this is also accompanied by lower editing efficiencies[5,48,49]. We selected two such proteins (SpCas9-HF1 and eSpCas9) due to their balanced activity and specificity as reported in literature[5,48,49], and compared their cellular editing rates (insertion/deletions, indels) with one of the enFnCas9 variants, en1. To assess the cellular genome editing potential of enFnCas9 variants, we constructed mammalian expression vectors which encode both the sgRNA and Cas9. Cas9 was fused to EGFP via a T2A linker, all under the control of an identical promoter, chicken β-actin (Fig. 3a). This design ensures uniform protein synthesis within cells and self-cleavage of the EGFP from the Cas9 by T2A. GFP-based fluorescence-activated cell sorting (FACS) was employed to ensure assay uniformity, followed by amplicon sequencing (Supplementary Fig. 1a, b).

We used 20-nt spacers containing gRNAs for which bona-fide off-targets were identified either through in silico prediction or GUIDE-seq[4,50]. Encouragingly, en1 showed higher editing rates than FnCas9 or SpCas9-HF1 and eSpCas9 variants at all the loci tested, without any detectable editing at the corresponding off-target sites (Fig. 3b).

As seen in our in vitro studies, the editing rate with enFnCas9 variants went up dramatically reaching ~90% at therapeutically relevant locus associated with Sickle cell anemia, *HBB* in HEK293T cells when combined with g21 (Fig. 3c). The activity of enFnCas9 variants on non-canonical PAMs (5'-NGR/NRG-3') observed in vitro prompted us to evaluate the genome editing efficiencies of these variants on such altered PAM-containing targets in human cells. Given its highest in vitro rate of DNA cleavage both at canonical 5'-NGG-3' and non-canonical 5'-NGA-3' PAMs, en31 was additionally examined for cellular genome editing on the targets with 5'-NGA/NAG-3' PAM (Fig. 3d, Supplementary Fig. 8a). Two GUIDE-seq validated gRNAs targeting 5'-NGA-3' PAM at *RUNX1* and *ZNF629* that had previously been reported[51] with highly promiscuous off-targets were investigated along with an additional 5'-NGA-3' PAM containing *FANCF1* site 2 gRNA. We confirmed robust editing at all three loci (~ 80% at *FANCF1*, ~60% at *RUNX1* and ~20% at *ZNF629*) (Fig. 3d). Expectedly, g21 was able to induce editing outcomes wherever g20 failed to do (*ZNF629*) (Fig. 3d). Remarkably, while previous reports had shown greater off-target editing than on-target activity with SpCas9 variant for *RUNX1* and *ZNF629* sites, we observed no off-target editing except for OT12 of *ZNF629* site (Fig. 3d). However, despite being identical to the on-target *ZNF629* site, off-targeting at OT12 was marginally detected for en31 compared with SpCas9 variant[51]. Furthermore, we also confirmed robust editing by en31 in one (~ 70% at *FANCF*) out of three sites having 5'-NAG-3' PAM with g21 (Supplementary Fig. 8a). Our results suggest that the PAM preference of en31 nuclease ranges from 5'-NGG > NGA > NAG-3' while retaining superior specificity of DNA interrogation even in the sites showing preponderance of off-targeting by high-fidelity SpCas9 variants.

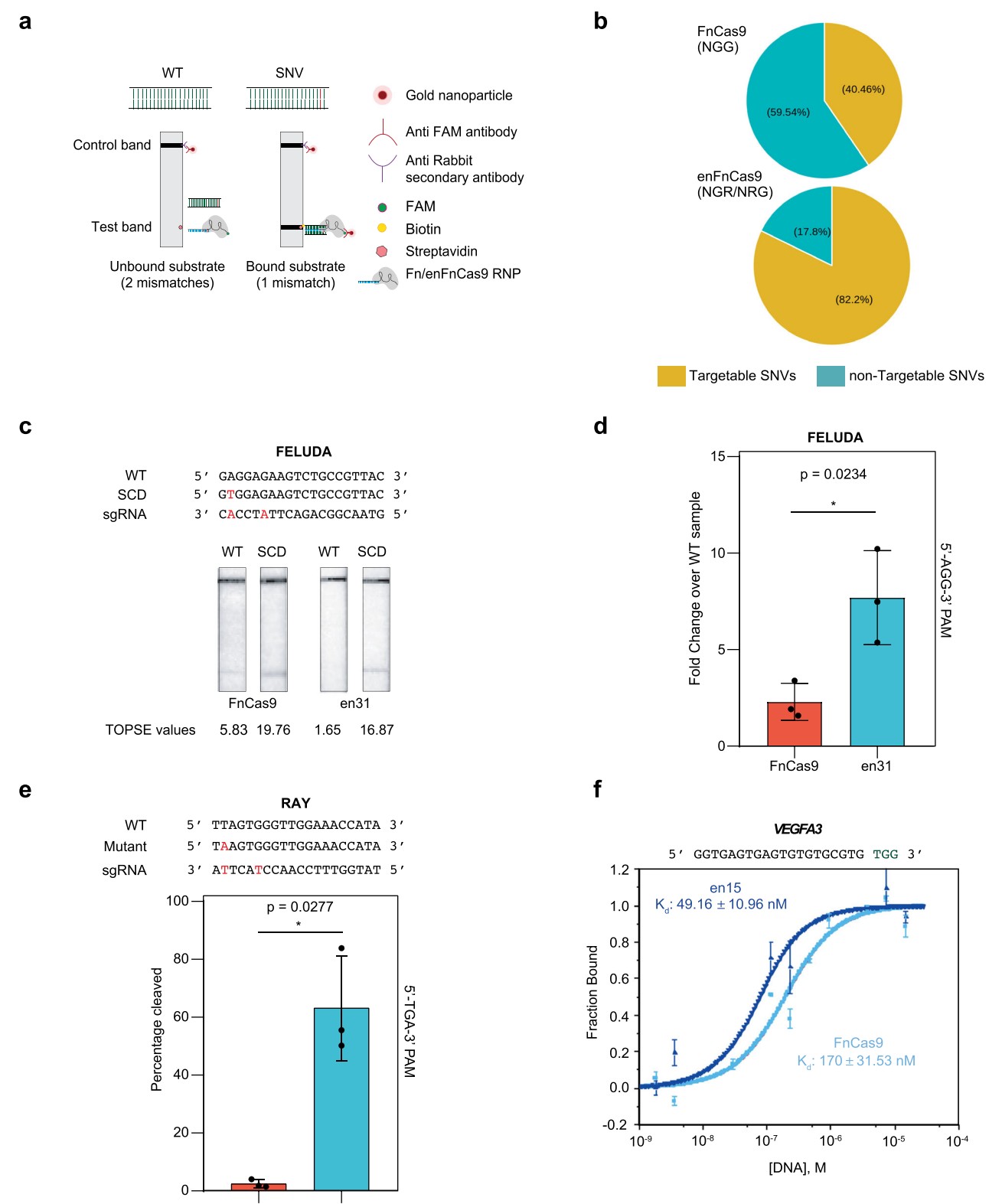

Finally, we speculated that higher editing outcomes by enFnCas9 variants might reflect in both higher NHEJ-mediated indels or HDR-mediated knock-in rates. Expectedly, we observed successful HDR-mediated knock-in of a long donor template (~6.8 kb) at the *DCX* locus in HEK293T cells for enFnCas9 variants as compared to FnCas9 (Fig. 3e, f). Importantly, en1 drastically outperformed SpCas9-HF1 and eSpCas9 in inducing HDR,

suggesting its suitability as a highly potent genome-editing protein (Fig. 3f).

## Single nucleobase specificity of enFnCas9 variants in human cells

Next, we investigated if the high editing efficiency and DNA binding affinity compromised the single-mismatch specificity of the enFnCas9

**Fig. 2 | enFnCas9 variant shows high sensitivity of SNV detection. a** Schematic representation showing the mode of SNV detection by FELUDA and RAY CRISPRDx platforms. The SNV is highlighted in red. Mismatch/es between DNA and RNA are mentioned w.r.t. sgRNA. **b** Pie chart showing the percentage of targetable and non-targetable SNVs by FnCas9 and enFnCas9 variants. **c** Representative image showing the outcome of lateral flow assay (LFA) for Sickle cell disease (SCD)-associated point mutation detection by FELUDA using FnCas9 and en31. WT and SCD target sequences are shown. The sickle cell mutation and FELUDA specific sgRNA with mismatches (PAM-proximal 2nd/6th, when counting away from PAM) are presented in red. Corresponding TOPSE values are given at the bottom. **d** Bar plot showing the quantification of SCD mutation discrimination by en31 and FnCas9 using FELUDA. Data is shown as fold enrichment over WT sample. Error bars represent mean ± SD of $n = 3$ independent experiments, and unpaired two-tailed Student's $t$-test was applied to calculate the $p$-value. **e** Bar plot showing the outcome of in vitro cleavage-based detection of SARS-CoV-2 N501Y by en31. The target sequences (Wild-type, WT, and mutant) and the sgRNA spacer sequence (PAM-proximal 2nd/6th mismatches, when counting away from PAM, are in red) are shown. The target mutation is highlighted in red. Error bars represent mean ± SD of $n = 3$ independent experiments, and unpaired two-tailed Welch's $t$-test was applied to calculate the $p$-value. **f** Microscale thermophoresis (MST) analysis showing the comparative binding affinity between FnCas9 and en15 on *VEGFA3* substrate DNA. Data is represented as a fraction bound RNP (y-axis) with respect to purified DNA substrate (Molar units M, x-axis). Error bars represent mean ± SD of $n = 3$ independent experiments. Source data are provided as a Source Data file.

variants. To this end, we interrogated the *FANCF* site 2 in HEK293T cells for which GUIDE-seq validated off-target with a single PAM-proximal mismatch was reported even by high-fidelity SpCas9 variants from independent studies[21,48,52]. Expectedly, we found comparable off-target editing (25% and 27%) as the on-target site (30% and 29%) by SpCas9-HF1 and eSpCas9, respectively (Fig. 4a). In sharp contrast, negligible (~1%) editing at the single mismatch off-target was observed for all the enFnCas9 variants when g20 was used, albeit with lower on-target editing (15-20% across the enFnCas9 variants) while FnCas9 did not induce substantial editing (~2%) (Fig. 4a). Interestingly, g21 or g22 increased the on-target editing efficiency up to 45% with en15 but no concomitant increase in off-targeting was seen (Fig. 4a). A similar trend was observed for both en1 and en31, although en1 showed a slight increase in off-target editing with a g21/22 (~8–10%), which was still three-fold lower than the high-fidelity SpCas9 variants tested (Fig. 4a). Taken together, these results underscore the combinatorial action of enFnCas9 variants and x-gRNAs for highly precise and robust editing.

## Robust cellular editing by enFnCas9 variants in multiple cell lines

Encouraged by robust and specific editing observed in HEK293T cells, we conducted a comparative analysis of on-target editing efficiencies between enFnCas9 variants, SpCas9 and its PAM-flexible variants such as SpCas9-NG[33] and SpRY[39]. We generated constructs in the same vector backbone to ensure uniform expression in cells (Fig. 3a).

We tested the on-target indel efficiencies in HEK293T cells, as well as in challenging-to-edit and therapeutically relevant induced pluripotent stem cells (iPSCs), and a retinal pigmented epithelial cell line (ARPE-19). Our results confirmed successful and comparable editing efficiencies (~80 to 90%) as SpCas9 for enFnCas9 variants (en1, en15, and en31) at the *FASN* locus in HEK293T cells, with en1 outperforming SpCas9 (Fig. 4b). Consistently, both en1 (18.6% indels) and en15 (24.3% indels) exhibited superior editing efficiencies in iPSCs at the *PAX6* locus as compared to SpCas9 (13.8%) (Fig. 4c). Similarly, we successfully knocked-out the *PAX6* gene in ARPE19 cells and confirmed the loss of protein expression by immunocytochemistry (Fig. 4d).

Furthermore, we compared the on-target editing rates of en1 with SpCas9, SpCas9-NG and SpRY at *HBB* and *EMX1* loci in HEK293T cells. Intriguingly, at *HBB* site, the indel efficiency of en1 (~45%) outperformed that of SpCas9 (~30%), SpCas9-NG (~40%) and SpRY (~30%) (Fig. 4e). Expectedly, en1 induced comparable editing efficiency (~60% indels) at the *EMX1* site as SpCas9, while both SpCas9-NG and SpRY achieved modest editing (~35% and ~20% indels, respectively) (Fig. 4f). These findings validate the superior on-target editing of enFnCas9 variants across multiple loci and different human cell types when compared to engineered PAM-flexible SpCas9 proteins.

## No detectable genome-wide off-targeting observed using enFnCas9 variants in human cells

Next, we explored the genome-wide target specificity of enFnCas9 variants, en1, en15 and en31 using the highly sensitive Digenome-seq which profiles off-target sites in an unbiased manner across the genome[53]. We compared these variants with SpCas9 and SpRY, at both the therapeutically relevant *HBB* site, and the commonly targeted *EMX1* locus for which on-target indel efficiencies were previously tested (Fig. 4e, f). In good agreement with our biochemical and targeted sequencing findings, enFnCas9 variants exhibited several orders of magnitude less off-targets at both the tested loci compared to SpCas9 (up to ~1300-fold less) and SpRY (up to ~700-fold less), despite achieving superior on-target editing outcomes (Fig. 5a, b; Supplementary Fig. 7a, b). Interestingly, we found 146 shared off-targets between SpCas9 and SpRY while for all the three enFnCas9 variants we found ≤2 shared off-targets when compared with both SpCas9 and SpRY in the *HBB* locus (Fig. 5c). Similar observation was persistent in the *EMX1* locus as well (Supplementary Fig. 7c). Furthermore, the targeted amplicon sequencing with a detection limit of 0.1% failed to capture any reads at the off-target sites identified by Digenome-seq at *HBB* locus. This underscores the remarkable DNA interrogation specificity of enFnCas9 in human cells while achieving robust on-target efficiency (Fig. 5d). Notably, the relaxed PAM-specificity of enFnCas9 variants and robust on-target activity does not confound genome-wide specificity in sharp contrast to the reported instances for SpRY[39] or other engineered orthologous Cas9s[54].

## en31 enables robust base editing in human cells

Despite the promises of Cas9 nuclease-based gene editing approaches, on-target genotoxicity combined with complex gene rearrangements has raised concerns about its use in therapeutic settings[55–59]. In contrast, the development of double-strand break (DSB)-free editing approaches such as base editing and prime editing has shown tremendous promise as safer alternatives[60]. Nevertheless, both the approaches suffer from guide-dependent off-targeting due to its reliance on enzymatically defective or inactive Cas9 for binding, an imperative feature for DSB free editing[61,62]. We sought to develop FnCas9/enFnCas9 base editors owing to its remarkable specificity of binding to cognate nucleobases in human cells (Supplementary Fig. 2b–d). Among the enFnCas9 variants, en31 showed the broadest PAM-flexibility and robust indel activity in human cells, which appeared to be an ideal candidate for evaluation as a base editor. To this end, we generated adenine base editor variants for FnCas9/en31 following previously reported ABEmax (ABE8.17dV106W) configurations which are shown to be highly efficient with improved gRNA-independent editing profiles—a feature important to curb the spurious RNA editing during base editing[63–65] (Fig. 6a, b). Given the larger share of ABE for pathogenic SNP correction[60], we characterized FnCas9-/en31-ABE for editing in human cells and compared it with SpNG-ABEmax8.17d, another PAM-flexible ABE variant that has been widely reported in literature[64]. We chose the therapeutically relevant sites of *HBG1/2* gene promoters responsible for hereditary persistence of fetal hemoglobin (HPFH)[66,67], a rare genetic condition known to ameliorate Sickle cell disease phenotype (Fig. 6c) and the commonly used *EMX1* site in HEK293T. For both loci, we observed low A > G substitution (1.7%/0.0% A6/A9 of −113/−116) and 3.7% A9 of *EMX1*) by en31-ABEmax8.17d with sg20 but drastically improved A > G substitutions

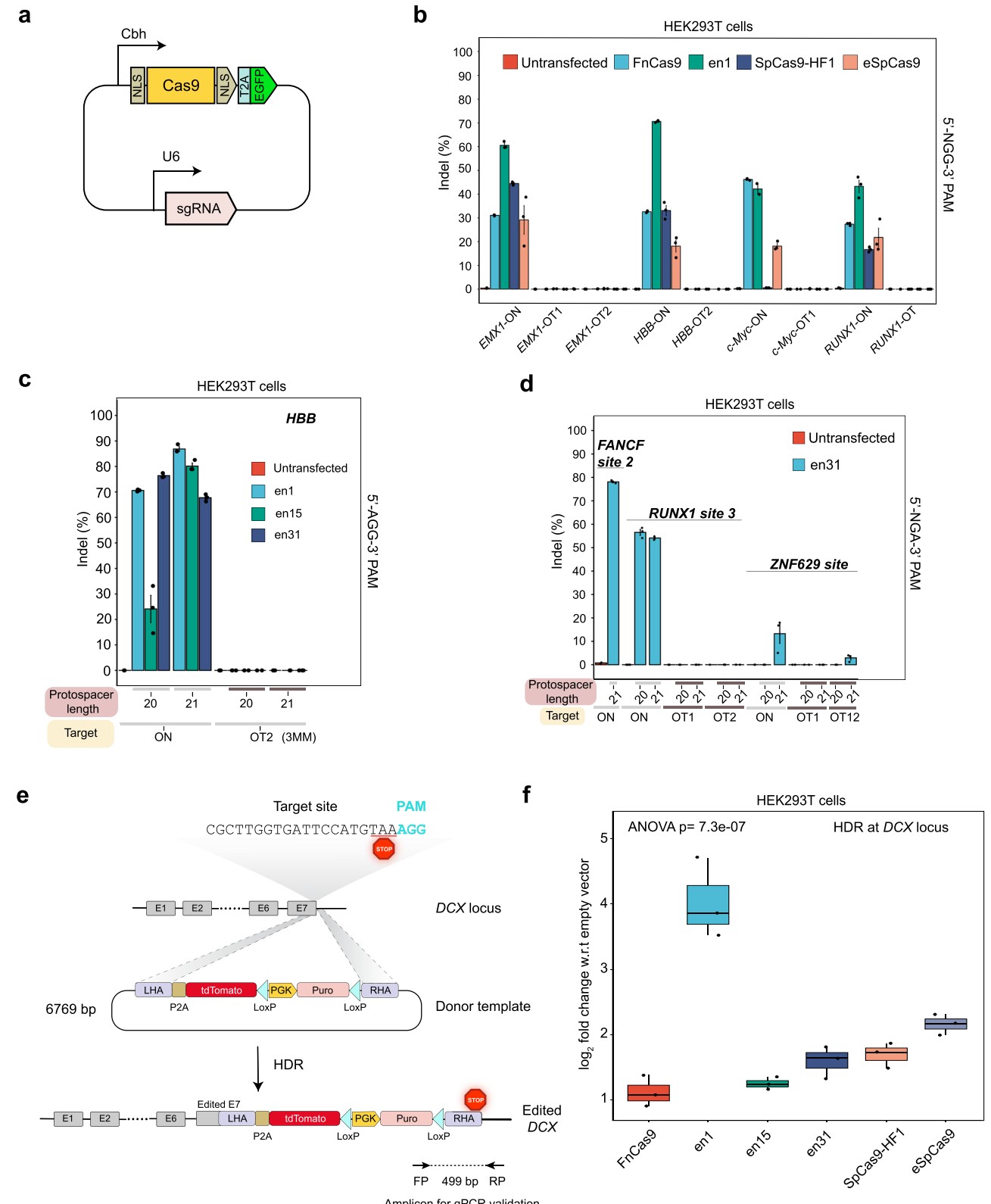

(14%/2.5% A6/A9 of −113/−116 and 3.7%/10.7%/13.33%/12.7% of A9, A12, A15 of *EMX1*) with sg21 (Fig. 6d, Supplementary Fig. 8b). Notably, SpNG-ABEmax8.17d showed reduced editing at both loci (6.7%/5.7% A6/A9 of −113/−116 and 0% of A12, A15 of *EMX1*) (Fig. 6d, Supplementary Fig. 8b). Expectedly, the off-targeting profile of en31-ABEmax8.17d in the validated off-target of *EMX1* (*EMX1-OT1*) was very low compared to SpNG-ABEmax8.17d (Supplementary Fig. 8b). However, FnCas9-

ABEmax8.17d did not install any appreciable A > G substitution over the baseline for both the tested loci (Fig. 6d, Supplementary Fig. 8b). We confirmed robust A > G substitution efficiency up to 72% with en31-ABEmax8.17d at different sites of the therapeutically relevant *HBG1/2* gene promoters (−111, −123/124, −175, −198 sites), with g21 outperforming g20 in all the sites tested (Fig. 6e, Supplementary Fig. 8c−e). Thus, en31-ABEmax8.17d with a g21 enabled robust base

**Fig. 3 | Superior cellular genome editing by enFnCas9 variants. a** Schematic showing the architecture of all-in-one plasmids expressing nucleases as T2A-EGFP fusion with gRNA cloning sites. **b** Bar plot showing the indel events (%) plotted on the Y-axis as obtained from amplicon sequencing upon targeting 5′-NGG-3′ PAM with sgRNA containing 20-nt spacer (g20) targeting *EMX1*, *HBB*, *c-Myc*, *RUNX1* loci, and its respective off-targets (OTs) by FnCas9, en1, SpCas9-HF1, and eSpCas9 in HEK293T cells. Untransfected cells were used as control. Error bars represent mean ± SEM of $n = 3$ independent biological replicates (except for *c-Myc* on-target of en1, $n = 2$) with individual values shown as dots. **c** Bar plot showing the indel events (%) plotted on the Y-axis as obtained from amplicon sequencing upon targeting 5′-AGG-3′ PAM containing *HBB* locus, and it's off-target site (OT2) by en1, en15, and en31 with sgRNAs either containing 20-nt spacer (g20) or 21-nt spacer (g21) in HEK293T cells. Untransfected cells were used as control. Error bars represent mean ± SEM of $n = 3$ independent biological replicates with individual values shown as dots. **d** Bar plot showing the indel events (%) plotted on the Y-axis

as obtained from amplicon sequencing upon targeting 5′-NGA-3′ PAM containing *FANCF1* site 2, *RUNX1* site 3 and *ZNF629* site loci and its respective off-targets by en31 with sgRNAs either containing 20-nt spacer (g20) or 21-nt spacer (g21) in HEK293T cells. Untransfected cells were used as control. Error bars represent mean ± SEM of $n = 3$ independent biological replicates with individual values shown as dots. **e** Schematic showing the design of the plasmid donor template for targeting *DCX1* locus in the HDR-mediated knock-in assay. **f** Box plot showing knock-in of a donor template at *DCX* locus by FnCas9, en1, en15, en31, SpCas9-HF1 and eSpCas9 in HEK293T cells. Data is represented as log2 fold change w.r.t. empty vectors, and analysed using one-way ANOVA, *p*-value is shown. The middle line within the box represents the median, the box edges represent the interquartile ranges and the whiskers indicate ± 1.5 × interquartile ranges of $n = 3$ independent biological replicates with individual values shown as dots. Source data are provided as a Source Data file.

editing in human cells with higher target base substitutions than SpNG-ABEmax8.17d in the tested loci.

## Tuning of base editing window by variable length gRNAs and en31-ABEmax8.17d

We speculated that widened PAM accessibility coupled with sx-gRNAs might offer distinct possibilities of base editing for en31-ABEmax8.17d, where conventional base editors might not be able to target the desired base. SpCas9 base editors (Sp-ABE8s) can only target bases that are within the targeting window of the deaminase (PAM-distal 3rd to 9th bases counting PAM at positions 21–23) because of protospacer length restrictions to 19/20-nt[64]. For editing other sites far away from the nearest available PAM, protein engineering to recognize a new PAM has been reported[23]. The en31-ABEmax8.17d protein showed a wider editing window with respect to the PAM (PAM-distal 3rd to 14th bases counting PAM at positions 22-24) when interrogating genomic sites with alternate adenine bases (Fig. 6f–i). Since, en31 can accommodate sx-gRNAs such as sg26 or sg28 (Supplementary Fig. 3b), we hypothesized that combining the two properties could facilitate the shifting of adenine base editing window to target inaccessible bases away from PAM. To validate this, we chose two loci in the human genome, *EMX1* (5′-GGG-3′ PAM) and *SERPINI1* (5′-TGA-3′ PAM), with a target base situated at inaccessible PAM-distal positions (3rd position for *EMX1* and 1st position for *SERPINI1*) (Fig. 6j, k; Supplementary Fig. 8f, g). Remarkably, by systematic modulation of gRNA lengths (g22-g26), we were successful in gradually shifting the editing window to the desired target while the target base editing in the primary window got serially diluted (Fig. 6j, k; Supplementary Fig. 8f, g). Thus, in principle, combining PAM-flexibility (5′-NGR/NRG-3′) and sx-gRNAs (up to 26) improves the target range of en31-ABEmax8.17d to 99.39% of all human G > A pathogenic SNVs identified in ClinVar[68,69] (Supplementary Fig. 9).

## Therapeutic base editing by en31-ABEmax8.17d restores protein expression in Leber congenital amaurosis 2 (LCA2) patient-specific retinal pigmented epithelial (RPE) cells

Finally, to establish the proof-of-concept validation for en31-ABEmax8.17d-based pathogenic mutation correction, we tested this protein for an ophthalmic condition namely, Leber congenital amaurosis, type 2 (LCA2). This is an early onset retinal dystrophy caused by mutations in *RPE65* gene. Mutation corrected, patient-specific iPSC-derived retinal pigmented epithelial (RPE) cell transplantations can be a viable therapeutic modality for the treatment of such inherited retinal dystrophies[70]. To this end, we isolated human dermal fibroblasts (HDFs) from the skin biopsy of a patient who was genotyped and reported to carry a G to A point mutation within the exon 9 of *RPE65* (c.992 G > A, p.Trp331Ter), leading to a premature stop codon (TGG > TAG) incorporation (Fig. 7a, b). Clinical examination of the patient by fundoscopy and optical coherence tomography

(OCT) has confirmed significant retinal thinning and attenuated photoreceptor cell layers[71]. The patient-specific HDFs were then reprogrammed to generate hiPSCs and further characterized for their morphology, genetic identity, stemness marker expression and pluripotency[72]. This patient-specific hiPSC line, LVPEIi005-A (LVIP02-LCA2-2) was transfected with en31-ABEmax8.17d and mutation specific sgRNAs. The NGS-based amplicon analysis has confirmed the successful installation of A > G substitution at the pathogenic mutation site in unsorted edited cell populations (~ 21% with sg21, 5′-AGG-3′ PAM) (Fig. 7c). Since, there were adjacent alternate PAM possibilities (5′-GGA-3′ and 5′-AAG-3′) at this locus, we also evaluated these target sites. We confirmed successful A > G base conversions (~ 13% for 5′-GGA-3′ and ~8% for 5′-AAG-3′), conforming to the earlier observations of en31 nuclease activity in the order of 5′-NGG > NGA > NAG-3′ (Fig. 7c). Importantly, two of the clonally expanded iPSC lines derived from the edited cells showed 100% A > G base conversion at the mutation position (A8), with very high target specificity and undetectable bystander edits at A10−12 base positions within the editing window (Fig. 7d).

The undifferentiated cells of the mutation corrected line, LVPEIi005-A-1 (LCA2-2-BE1) maintained the pluripotency and expressed the major stem cell markers (Supplementary Fig. 10a, b) and upon random differentiation, the day 10 embryoid bodies (d10-EB) are comprised of cells that expressed all three lineage markers (Supplementary Fig. 10c). Moreover, the karyotypes of both the patient-specific line (LCA2-2) and the mutation corrected, patient-specific iPSC line (LCA2-2-BE1) were found to be normal (Supplementary Fig. 10d, e). Further, we differentiated the healthy control iPSC line, LVPEIi001-B (LVIP02-NC-F2-1), the patient-specific iPSC line, LVPEIi005-A (LVIP02-LCA2-2) and the mutation corrected patient-specific iPSC line, LVPEIi005-A-1 (LCA2-2-BE1) into eye fields and generated pure cultures of mature RPE cells and neuro-retinal organoids[73–75]. Upon eye lineage differentiation, all three lines followed the normal timelines and formed the typical self-organized, eye field primordial (EFP) clusters precisely within 3-4 weeks of differentiation induction. The EFPs displayed the centrally positioned neuro-retinal islands, with the migrating retinal pigmented epithelial (RPE) progenitors all around their margins (Supplementary Fig. 10f). The 3D retinal organoids and the enriched RPE cultures derived from EFPs were further characterized for the expression of retinal lineage-specific markers. At d45, the RPE progenitors of all the three lines expressed the transcripts of lineage commitment markers such as *PAX6, MITF, CRALBP, MERTK, TYR* at comparable levels (Supplementary Fig. 10h). Expectedly, the *RPE65* mRNA level was barely detected in patient iPSC-RPE cells (Supplementary Fig. 10h). At d45, the RPE progenitors appear non-pigmented (Fig. 7e i–iii) and they gradually mature and acquire pigmentation in culture. Interestingly, the healthy control (F2) and the mutation corrected patient-line (LCA2-2-BE1) acquired comparable levels of pigmentation at d75 (Fig. 7e iv, vi). However, the patient line (LCA2-2)

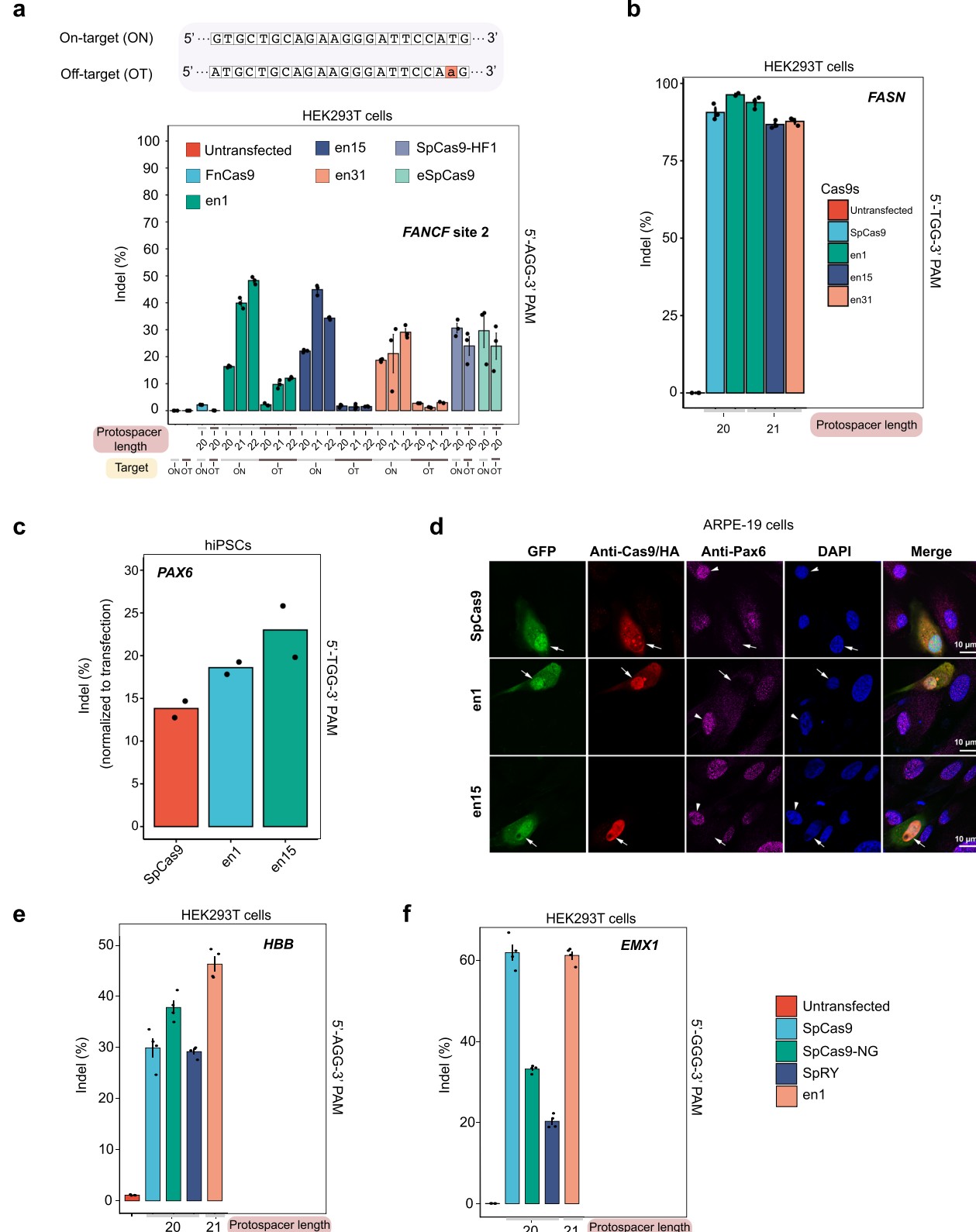

showed a slight delay in pigment accumulation (Fig. 7e v). It started showing accumulation of pigmentation only around d90 of differentiation (Supplementary Fig. 10g). The mature RPE cells at d75 established tight junctions and displayed a cobblestone morphology, as marked by the Phalloidin-FITC staining (Fig. 7f i–iii). We further checked for the expression of RPE-specific proteins, MITF and RPE65 by immunolabeling of d75 RPE cultures. While all three lines expressed comparable levels of MITF (Fig. 7f iv–vi), the RPE65 expression was undetectable in patient-specific cells (LCA2-2-RPE) (Fig. 7f viii). Encouragingly, RPE65 protein expression was restored in mutation-corrected patient cells (LCA2-2-BE1-RPE) to a level comparable to that of healthy control cells (F2_RPE) (Fig. 7f vii, ix). We further confirmed the restoration of RPE65 expression by western blotting using a polyclonal antibody and observed that the full-length RPE65 protein

**Fig. 4 | Single nucleobase specificity and robust cellular editing by enFnCas9 variants in multiple human cell lines. a** Bar plot showing the indel events (%) plotted on the Y-axis as obtained from amplicon sequencing upon targeting 5′-AGG-3′ PAM containing *FANCF1* site 2, and its off-target site (OT) by FnCas9, en1, en15, en31, SpCas9-HF1, and eSpCas9 with sgRNAs containing 20/21/22-nt spacers (g20, g21 and g22 respectively) in HEK293T cells. Single nucleotide mismatch of the off-target site (OT) is shown in lowercase and highlighted in red. **b** Bar plot showing the indel events (%) plotted on the Y-axis as obtained from amplicon sequencing upon targeting 5′-TGG-3′ PAM containing *FASN* locus by SpCas9, en1, en15 and en31 with sgRNAs either containing 20-nt spacer or 21-nt spacer in HEK293T cells. **a, b** Untransfected cells were used as control. Error bars represent mean ± SEM of $n = 3$ independent biological replicates with individual values shown as dots. **c** Bar plot showing the indel events (%) normalized to the transfection efficiency as obtained from T7E1 assay upon targeting *PAX6* locus by SpCas9, en1 and en15 in

hiPSC cells. Values represent the mean of $n = 2$ independent biological replicates. **d** Representative immunofluorescence images showing the transfected ARPE-19 cells expressing the GFP reporter (in green) and stained with anti-HA/anti-Cas9 (in red), anti-PAX6 (in magenta) and counterstained with DAPI (in blue), upon targeting *PAX6* locus using SpCas9, en1 and en15. Merged panel depicts all the cells present in the field. Cas9-expressing cells that lost PAX6 expression are marked by arrows, and the untransfected cells that express normal levels of PAX6 are marked by arrowheads (n = 3). Scale bars: 10 μm **e**, **f** Bar plots showing the indel events (%) from unsorted cells, plotted on the Y-axis as obtained from amplicon sequencing upon targeting 5′-NGG-3′ PAM containing *HBB* and *EMX1* loci respectively by SpCas9, SpCas9-NG, SpRY with sgRNA containing 20-nt spacer, and en1 with sgRNA containing 21-nt spacer. Untransfected cells were used as control. Error bars represent mean ± SEM of $n = 4$ independent biological replicates with individual values shown as dots. Source data are provided as a Source Data file.

expression is rescued in corrected patient cells, at levels comparable to that of the healthy control cells (Fig. 7g). However, the unedited patient cells expressed only the truncated form of the protein, which was not detected by the monoclonal antibody used in immunostaining experiments (Fig. 7f, g). Similarly, the neuro-retinal organoids of all three lines at d35, expressed the neuro-retina specific transcripts, *PAX6, SIX6, OTX2, CHX10, NEUROD1, CALB1, RCVRN, ARR3* and *RHO* (Supplementary Fig. 10i).

Our results have confirmed that the patient-specific iPSC line has retained the ability to form the neuro-retina and mature RPE cells in vitro. Albeit, it exhibited a slight delay in RPE cell maturation and pigmentation, and lacked the expression of normal levels of *RPE65* mRNA and full-length proteins. However, successful editing of the mutations in both the alleles has completely restored the normal levels of mRNA and full-length protein expressions, thus confirming the reversal of disease phenotype in vitro.

Taken together, en31-ABEmax8.17d can be successfully utilised for robust and precise nucleobase correction, with undetectable bystander edits and genotoxicity in therapeutic conditions. This leads to a complete reversal of disease-associated phenotypes in edited retinal cells.

## Discussion

In the present study, we have developed enFnCas9 variants, and demonstrated the efficacy and specificity of these variants in targeted genome editing and its applications in therapeutics and diagnostics. Interestingly, the specificity of these variants appears to stem from the DNA interaction properties of FnCas9 independent of the engineered residues at the PAM-duplex region sandwiched by WED-PI domains of the enzyme. Thus, we observed that even after substantially improving DNA binding affinity and activity, enFnCas9 variants showed minimal to undetectable editing at GUIDE-Seq validated off-targets and at the genome-wide level. This is in sharp contrast to both the high-fidelity and PAM-flexible versions of SpCas9 such as eSpCas9[49], SpCas9-HF1[48], SpCas9-NG[33] and SpRY[39], and numerous other orthologs of both natural and engineered Cas9s reported in literature[12,54]. These results show that enFnCas9 variants possibly negotiate off-targets through a different mechanism than SpCas9 proteins and its variants[76].

Furthermore, we purified the recombinant proteins, and compared the activity and specificity of two of the enFnCas9 variants (en1 and en31) with the recently developed SpRY[39], a most PAM-flexible engineered variant and Superfi-Cas9[77], a next-generation high-fidelity engineered Cas9 variant (Supplementary Fig. 11a, b). Using in vitro assays, we observed that SpRY and Superfi-Cas9 exhibited slower rate of DNA cleavage compared to enFnCas9 variants, which explains the markedly decreased cellular editing by SpRY as shown in this study, and by Superfi-Cas9 as reported recently[78,79] (Supplementary Fig. 11c). Notably, enFnCas9 variants showed similar kinetics as SpCas9 which explains the robust cellular activity of these variants (Supplementary Fig. 11c). Next, we evaluated the single mismatch specificity of high-

fidelity Superfi-Cas9 at the PAM-distal stretch of the protospacer (which serves as a checkpoint of DNA interrogation fidelity which guides nuclease activation) using mismatch walking assay. Interestingly, we found around 0.5-fold lower PAM-distal single mismatch specificity of Superfi-Cas9 compared with en31 (no detected DNA cleavage), an attribute of superior specificity of enFnCas9 variants observed in our study (Supplementary Fig. 11d). Moreover, a careful scrutiny of editing efficiency on *FANCF* site 2 by one of the enFnCas9 variants, en15, suggests drastically improved on-target to off-target ratio (~45% on-target editing and ~1.4% off-target editing, ~32-fold) of enFnCas9 variant than Superfi-Cas9 (2852 on-target reads and 500 off-target reads, ~6-fold)[78]. This suggests superior efficiency and specificity compared to SpCas9 and its engineered variants described in literature so far (Fig. 4a).

Additionally, we constructed a truncated en1 by deleting its REC2 domain (ΔS112-A297, which does not show any tertiary contacts with the rest of the protein) and reduced the size of en1 to ~170 kDa, closer to that of SpCas9 (~159 kDa) (Supplementary Fig. 11e). Remarkably, en1-ΔREC2 retains its DNA cleavage efficiency at a level similar to en1, while we observed reduction in cleavage efficiency with FnCas9-ΔREC2 (Supplementary Fig. 11f). This is in sharp contrast to SpCas9-ΔREC2 where the substantial loss of activity was seen upon deletion[1]. These findings hint at the distinct domain modularity of FnCas9, which is different from SpCas9.

In this study, we have used ABE8.17dV106W adenine deaminase variant for en31-ABEmax carrying V106W mutation to mitigate the guide-independent off-targeting impacting global transcriptome, which has been a concern in recent times for the use of base editors without compromising the A > G substitution efficiency of ABEs[64,65,80,81]. Furthermore, we employed the classic dual TadA architecture with wild-type TadA followed by mutated TadA (TadA*) to generate en31-ABE8.17max. Recent reports have shown dimeric TadAs can be replaced with monomeric TadA* without sacrificing editing efficiency[64,80,81]. Therefore, a monomeric TadA*-containing en31-ABEmax can be developed, resulting in a reduced construct size of ~600 bp (including the linker), which corresponds to a reduction of ~20 kDa in protein size. Additionally, combining such a system with mini-enFnCas9 might prove valuable. Nonetheless, more processive deaminase variants (ABE8e[81] and ABE8.20[64]) or the deaminase variant with a narrow editing window (ABE9[82]) can be used with en31 variant to develop more potent enFnCas9-based base editors. These approaches address scenarios where the en31-ABEmax8.17d variant either failed to achieve robust editing or to meet specific therapeutic case scenarios.

The ability to extend gRNA length allows a wide targeting range and base editing scope across the genome with enFnCas9 variants. This property of tuning the base editing window without the need for PAM engineering is a distinct outcome of this study. It should be noted that Cas9 variants such as SpRY, that are not limited by PAM constraints, have been reported in literature and have been applied in genome editing[35]. However, a recent biophysical study highlights

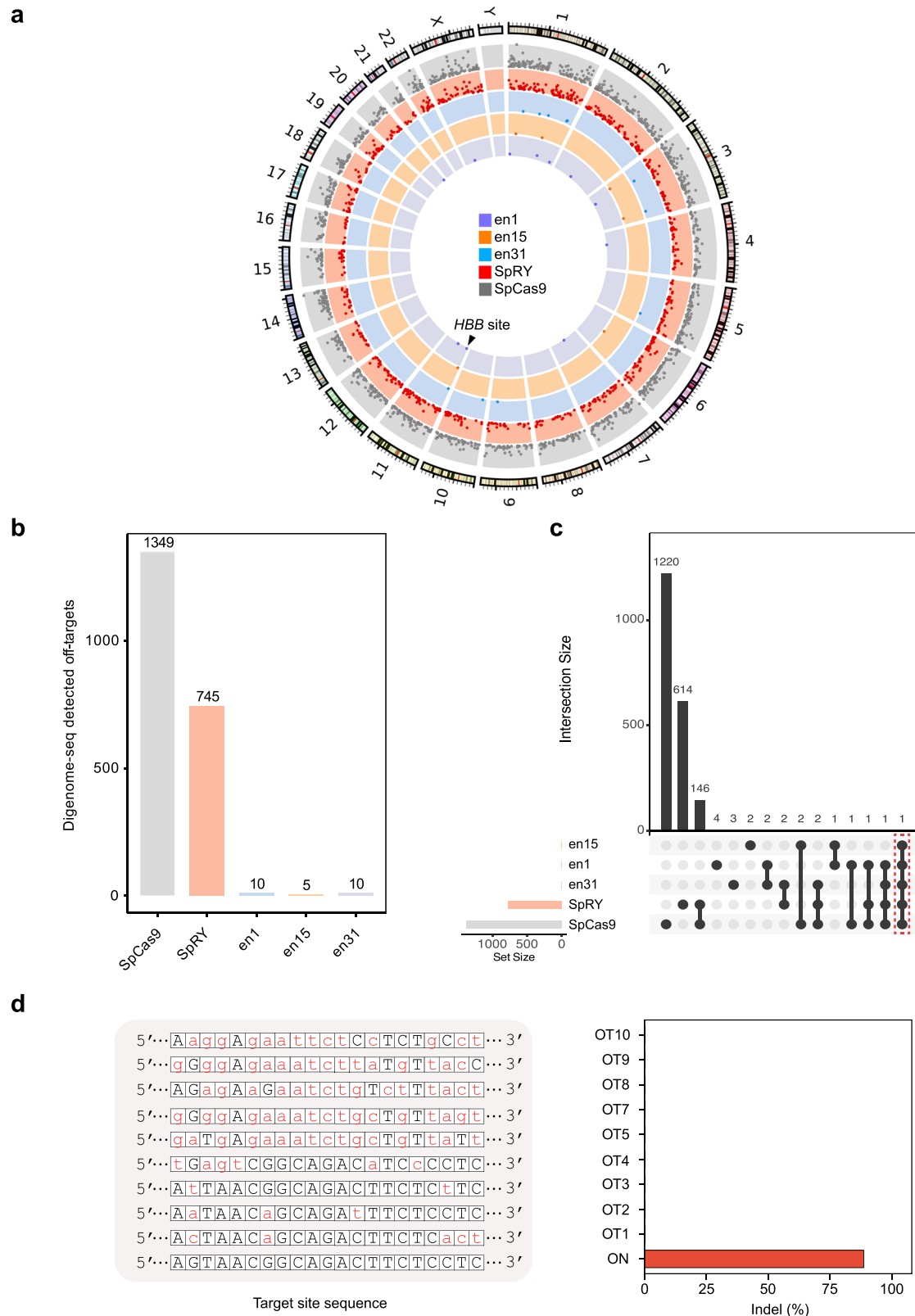

-1000-fold slower kinetics of SpRY as compared to SpCas9 and the possibility of off-target binding[83]. In therapeutic situations where rapid editing and clearance of the editor from cells is desirable, enFnCas9 variants might prove beneficial due to the combination of high specificity, robust activity, and targeting flexibility. Our results indicate that engineering residues that regulate PAM-duplex contacts in the Cas9 backbone can significantly improve editing efficiency without affecting specificity. This strategy can potentially be extended to other orthologous Cas systems that possess higher intrinsic specificity but have low cellular activity. Furthermore, it is worth noting that our engineering strategy breaks the notion of the trade-off involved between activity and specificity where specificity has been achieved by sacrificing the efficiency of the enzyme even on the canonical PAM upon engineering[5,7].

**Fig. 5 | No detectable genome-wide off-targeting by enFnCas9 variants in human cells. a** Circos plot showing the comparative off-targeting profile by SpCas9 and SpRY programmed with 20-nt spacer containing sgRNA, and en1, en15, and en31 programmed with 21-nt spacer containing sgRNA against *HBB* locus in HEK293T cells, as captured by Digenome-seq. The scatter dots indicate the on-target site (marked by arrowhead) and off-target sites (*n* = 1). Each concentric circle corresponds to each Cas9 variant, as labeled in the figure. **b** Bar plot showing the number of off-target sites plotted on the Y-axis from Digenome-seq assay for

SpCas9, SpRY, en1, en15, and en31 at *HBB* locus in HEK293T cells. **c** UpSet plot showing the hits identified in the Digenome-seq at *HBB* locus across Cas9 variants. The on-target is outlined with a dotted red rectangle. **d** Bar plot showing the indel events (%) at *HBB* locus by en1 plotted on the Y-axis as obtained from amplicon sequencing of the Digenome-seq detected off-targets (OTs) in HEK293T cells as compared to the on-target (ON) (*n* = 1). The mismatches of the off-targets are shown in lowercase and highlighted in red. OT6 was a drop-out due to PCR failure. Source data are provided as a Source Data file.

Based on our studies, we recommend using the en1 variant for general cellular editing assays with a 5′-NGG-3′ PAM, en15 for high-fidelity editing with a 5′-NGG-3′ PAM, and en31 for nuclease and base editing with altered PAMs (5′-NRG/NGR-3′). The en31 variant coupled with sx-gRNA (more than 24-nt long spacer) is recommended when editing window modulation for base editing is warranted. Notably, compatibility of enFnCas9 variants with sx-gRNAs provides alternatives for enhancing specificity when targeting highly promiscuous repetitive regions in the genome, given the natural rarity of occurrence of perfectly or near-perfectly matched off-targets with longer lengths of matched spacers. Thus, we propose coupling of super-specific enFnCas9 variants with sx-gRNAs to provide efficient yet safe editing outcomes in the difficult regions of the genome. However, this claim demands further experimental validation.

Altogether, the enFnCas9 variants hold a lot of promise for safe and efficient nuclease-mediated genome editing. They also present potentially attractive avenues for double-strand break-free editing (such as base and prime editors), especially since recently reported prime editing by FnCas9 has suffered from low activity[84].

## Methods

### Ethical statement

The overall study was reviewed and approved by the Institutional Review Board at the CSIR-IGIB, New Delhi, India. All experiments involving human biological samples and human iPSC lines followed the tenets of the Declaration of Helsinki and were carried out aseptically, in adherence to the standard laboratory practices, ethical and biosafety guidelines. The study was reviewed and approved by institutional regulatory bodies such as the Institutional Ethics Committee (IRB/IEC Ref. No: 08011; 02-16-003), Institutional Committee for Stem Cell Research (IC-SCR Ref. No. 04-15-016; 02-17-001) and Institutional Bio-Safety Committee (IBSC Ref. No. 06-21-009; 06-22-012) at the LV Prasad Eye Institute (LVPEI), Hyderabad, India. All human samples were collected after duly explained and written informed consent from the patient volunteer and their parent/guardian. No monetary compensation has been paid to any of the study volunteers.

### Plasmid construction

Point mutations and deletions were done by inverse PCR method on FnCas9 cloned in pE-SUMO vector backbone (LifeSensors) where intended changes were made on the forward primer, and the entire plasmid was amplified by inverse PCR. Point mutations on PX458-3xHA-FnCas9 and PX458-3xHA-SpCas9 vectors were done to generate catalytically inactive (dead) double mutants for ChIP-seq assay. Point mutations on the pET-His6-dFnCas9GFP and PX458-3xHA-FnCas9 (Addgene130969) were done to generate respective en1, en15, and en31 constructs by essentially following the method described earlier[4]. Base editing constructs (PX458-3xHA-FnCas9A-BEmax8.17d, PX458-3xHA-en31FnCas9ABEmax8.17d, and PX458-3xHA-SpCas9-NG-ABEmax8.17d) were synthesized as gene blocks (GenScript) and subcloned in modified PX458-3xHA-FnCas9 and modified PX458-3xHA-SpCas9 (with a unique EcoRI site generated by site-directed mutagenesis) in AgeI and EcoRI sites using In-Fusion ® HD Cloning Kit. PX458-3xHA-SpCas9-NG and PX458-3xHA-SpRY were generated in the PX458-3xHA-SpCas9 (Addgene130968)

backbone using In-Fusion ® HD Cloning Kit. gRNAs were cloned in the BbsI sites of PX458-3xHA-FnCas9, PX458-3xHA-en1FnCas9, PX458-3xHA-en15FnCas9, PX458-3xHA-en31FnCas9, PX458-3xHA-SpCas9-HF1, eSpCas9(1.1) (Addgene 71814), PX458-3xHA-SpCas9-NG, PX458-3xHA-SpCas9-RY, PX458-3xHA-FnCas9ABEmax8.17d, PX458-3xHA-en31FnCas9ABEmax8.17d, and PX458-3xHA-SpCas9-NG-ABE-max8.17d constructs for cellular genome editing and ChIP-seq assays by essentially following the method described earlier[85]. All of the constructs were sequenced before being used. Plasmids used in this study are available from Addgene (see Supplementary Table 1 for details). The Superfi-Cas9 plasmid was a kind gift from the lab of David Taylor (University of Texas).

### Protein purification and sgRNA purification

The proteins used in this study were purified as reported previously[4,33]. Briefly, plasmids for Cas9 from *Francisella novicida* were expressed in Escherichia coli Rosetta2 (DE3) (Novagen). The protein-expressing Rosetta2 (DE3) cells were cultured at 37 °C in LB medium (supplemented with 50 mg/mL kanamycin) until $OD_{600}$ reached 0.6 and protein expression was induced by the addition of 0.5 mM isopropyl-β-D-thiogalactopyranoside (IPTG). The Rosetta2 (DE3) cells were further cultured at 18 °C overnight and harvested by centrifugation. The *E. coli* cells were resuspended in buffer A (20 mM Tris-HCl, pH 8.0, 20 mM imidazole, and 1 M NaCl), and lysed by sonication, and centrifuged. The lysate was mixed with Ni-NTA beads (Roche), the mixture was loaded into a Poly-Prep Column (BioRad), and the protein was eluted by buffer B (20 mM Tris-HCl, pH 8.0, 0.3 M imidazole, and 0.3 M NaCl). The affinity eluted protein was mixed with ion-exchange beads (SP Sepharose Fast Flow, GE Healthcare) equilibrated with buffer C (20 mM Tris-HCl, pH 8.0, and 0.15 M NaCl) and the protein was eluted by buffer D (20 mM Tris-HCl, pH 8.0, and 1 M NaCl). SpCas9, SpRY, Superfi-Cas9, AaCas12b and Cas14a1 and were purified essentially by following the purification methods described earlier with some modifications[4,79,86,87]. The concentration of purified proteins was measured by the Pierce BCA protein assay kit (Thermo Fisher Scientific). The purified proteins were stored at −80 °C until further use.

In vitro transcribed sgRNAs were synthesized using MegaScript T7 Transcription kit (Thermo Fisher Scientific) using T7 promoter containing template as substrates. IVT reactions were incubated overnight at 37 °C followed by NucAway spin column (Thermo Fisher Scientific) purification as described earlier[4]. IVT sgRNAs were stored at -20 °C until further use.

### In vitro cleavage (IVC) assay

For the DNA cleavage study, PCR linearized pUC119 plasmid containing the target sequence and the respective PAM (mentioned in the respective legends) was used as the substrate for in vitro cleavage experiments. The linearized pUC119 plasmid (50 ng or ~5 nM) was incubated at 37 °C for 0.5–5 min with the Cas9-sgRNA complex (50 nM) in 10 μL of reaction buffer, containing 20 mM HEPES, pH 7.5, 150 mM KCl, 10 mM $MgCl_2$, 1 mM DTT, and 5% glycerol. The reaction was stopped by the addition of a quenching buffer, containing EDTA (20 mM final) and Proteinase K (40 ng). The reaction products were resolved, visualized, and quantified with a MultiNA microchip electrophoresis device (SHIMADZU)[33].

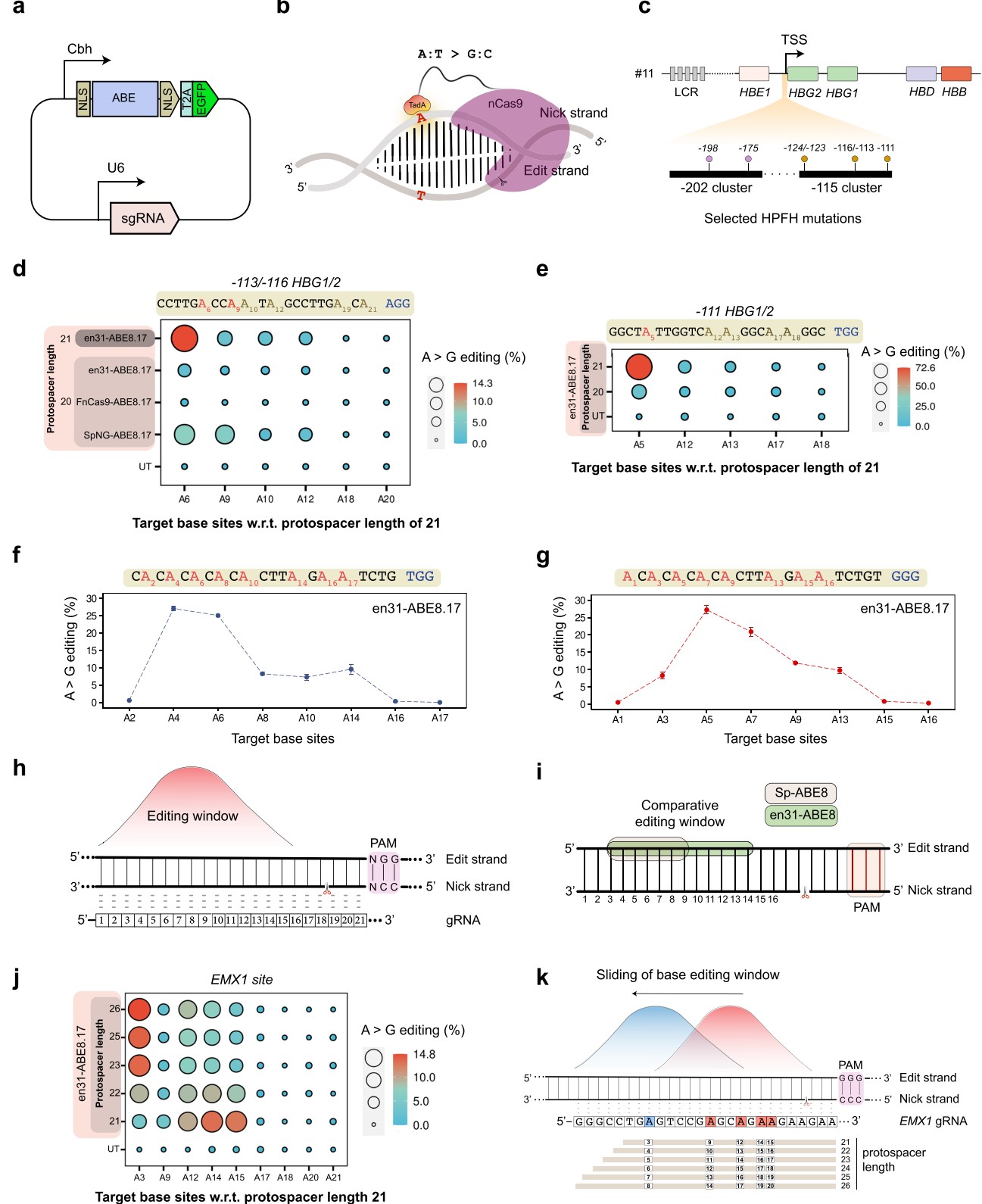

The rest of the IVC assays were done as described earlier[4]. Details of substrates, concentrations, and incubation time are mentioned in the respective figure legends.

Quantification of IVC assays from agarose electrophoresis was done using ImageJ and the kinetics data were fitted with a one-phase exponential association curve using Graphpad Prism.

The in vitro cleavage percentage was calculated using the following formula:

$$\text{Percent cleavage} = [(\text{Intensity of cleaved product})/ (\text{Intensities of cleaved and uncleaved products})] \times 100$$

**Fig. 6 | en31-ABEmax8.17d enables robust and flexible base editing in human cells. a** Schematic showing the architecture of all-in-one plasmids expressing adenine base editors as T2A-EGFP fusion with gRNA cloning sites. **b** Schematic showing the mode of the action of an adenine base editor (ABE). The nick position by ABE is indicated by a scissor on the nick strand. Target base is on the edit strand. **c** Schematic showing the architecture of the gamma-globin promoter highlighting the point mutations responsible for Hereditary persistence of fetal hemoglobin (HPFH) condition. Two clusters of point mutations located at 200 and 115 nucleotides from the transcription start site (TSS) are zoomed in. **d, e** Ballon plots showing the A to G editing events (%) as obtained from amplicon sequencing upon targeting *-113/-116* (**d**) *and -111* (**e**) sites of *HBG1/2* promoter by FnCas9-ABEmax8.17d, en31-ABEmax8.17d, and SpNG-ABEmax8.17d with sgRNA containing 20/21-nt protospacer (g20 and g21 respectively) in HEK293T cells. Target bases for adenine base editing are highlighted in red and numbered w.r.t. g21. **f, g** Line plot showing A to G editing events (%) plotted on the Y-axis obtained from amplicon

sequencing upon targeting endogenous loci having alternately present target 'A' bases by en31-ABEmax8.17d in HEK293T cells. Target bases (As) for adenine base editing are numbered on the X-axis and marked red. Error bars represent mean ± SEM of *n* = 3 independent biological replicates. **h** Schematic showing the adenine base editing window of en31-ABEmax8.17d. Apex of the bell curve indicates the optimal editing window. Numbers indicate the base positions across the protospacer of the gRNA. **i** Schematic showing the comparative base editing window between Sp-ABEmax8.17d (shaded in pink) and en31-ABEmax8.17d (shaded in green). **j** Balloon plot showing modulation of the base editing window by en31-ABEmax8.17d with gRNAs of varying spacer lengths (g21−26) at *EMX1* locus in HEK293T cells. **k** Schematic showing the sliding of the base editing window from primary window (shown in red) to secondary window (shown in blue) by x-/sx-gRNAs. **d, e, j** Untransfected cells were used as control. Values represent the mean of *n* = 3 independent biological replicates. Source data are provided as a Source Data file.

## PAM discovery assay

The PAM discovery assays were performed, as previously described[33]. Briefly, a library of pUC119 plasmids containing eight randomized nucleotides downstream of the target sequence was incubated at 37 °C for 5 min with the FnCas9−sgRNA complex (50 nM), in 50 µL of the reaction buffer. The reactions were quenched by the addition of Proteinase K and then purified using a Wizard DNA Clean-Up System (Promega). The purified DNA samples were amplified for 25 cycles, using primers containing common adapter sequences. After column purification, each PCR product (~ 5 ng) was subjected to the second round of PCR for 15 cycles, to add custom Illumina TruSeq adapters and sample indices. The sequencing libraries were quantified by qPCR (KAPA Biosystems) and then subjected to paired-end sequencing on a MiSeq sequencer (Illumina) with 20% PhiX spike-in (Illumina). The sequencing reads were demultiplexed by primer sequences and sample indices, using NCBI Blast+ (version 2.8.1) with the blastn-short option. For each sequencing sample, the number of reads for every possible 8-nt PAM sequence pattern ($4^8$ = 65,536 patterns in total) was counted and normalized by the total number of reads in each sample. For a given PAM sequence, the enrichment score was calculated as log2-fold enrichment as compared to the untreated sample. PAM sequences with enrichment scores of − 2.0 were used to generate the PAM wheel using KronaTools (v2.7) (https://hpc.nih.gov/apps/kronatools.html) and the sequence logo representation using WebLogo 3 (http://weblogo.threeplusone.com/create.cgi).

## en/FnCas9 based SNP detection

### (i) in vitro *cleavage (IVC) assay*

The RNA substrates were reverse transcribed into cDNA (Qiagen), followed by PCR amplification or the DNA substrates were only PCR amplified (Invitrogen) and further purified. The substrates were treated with a pre-assembled 500 nM en/FnCas9-sgRNA (1:1) RNP complex in a tube containing reaction buffer (20 mM HEPES, pH 7.5, 150 mM KCl, 1 mM DTT, 10% glycerol, 10 mM MgCl₂) at 37 °C for 10 min. The reaction was inactivated by using 1 µL of Proteinase K (Ambion) at 55 °C for 10 min, followed by the removal of residual gRNA by RNase A (Purelink) at 37 °C for 10 min. The cleaved products were visualized on a 2% agarose gel and quantified.

### (ii) via *lateral flow assay*

5' biotin-labeled amplicons were treated with reconstituted en/FnCas9 RNP complex (prepared by equimolar mixing 3' FAM labeled-Chimeric gRNA and en/FnCas9 in a buffer containing 20 mM HEPES, pH7.5, 150 mM KCl, 1 mM DTT, 10% glycerol, 10 mM MgCl₂ and rested for 10 min at RT) for 10 min at 37 °C. Wherever active en/FnCas9 was used, MgCl₂ was omitted from the buffer for making it catalytically inactive. After incubation, an 80 µL Dipstick buffer was added to the reaction

tube followed by the addition of one Milenia HybriDetect lateral flow strip and kept for 2−5 min at RT to observe test and control bands. Further background-corrected band intensity values were calculated through a smartphone application, True Outcome Predicted via Strip Evaluation (TOPSE)[40,41].

## Fluorescence assay (dFnCas9)

250 nM biotin-labeled PCR amplicons carrying 580 bp long SARS-CoV-2 region with N501Y mutation were used for attaching DNA substrate to the wells of streptavidin-coated plate by 10 min incubation at room temperature. Wells were rinsed thrice with the wash buffer to get rid of the unbound amplicons (25 mM Tris-Cl, pH 7.2; 300 mM NaCl; 0.1% BSA, 0.05% Tween®-20 Detergent) before using for the binding assay. dFnCas9-GFP RNP complex was pre-assembled in the binding buffer (20 mM HEPES, pH 7.5, 150 mM KCl, 1 mM DTT, 10 mM MgCl₂) by incubating 200 nM dFnCas9-GFP with 200 nM sgRNA for 10 min at room temperature. Reaction was initiated by adding pre-assembled RNP to the wells of 96-well streptavidin coated plate (Thermo Fisher Scientific; Cat 15119) pre-attached with biotin labeled amplicons and incubated at 37 °C for 10 min. Fluorescence was measured using a fluorescence plate reader ($\lambda_{ex}$: 485 nm; $\lambda_{em}$: 528 nm, transmission gain: optimal) (Tecan Infinite Pro F200).

## Fluorescence assay (AaCas12b and Cas14a1)

AaCas12b and Cas14a1 RNP complexes were pre-assembled by incubating 200 nM AaCas12b and Cas14a1 with 200 nM respective sgRNA for 10 min at room temperature. Reaction was initiated by adding pre-assembled RNP, 20 nM ssDNA activator, 100 ng background genomic DNA and 200 nM custom synthesized homopolymer ssDNA FQ reporter as described earlier[43,88] (GenScript) in a cleavage buffer (40 mM Tris-HCl, pH 7.5, 60 mM NaCl, 6 mM MgCl₂). The reaction was incubated in a 96-well flat bottom clear, black polystyrene microplate (Corning, cat no. CLS3603) at 37 °C up to 180 min with fluorescent measurements taken every 10 min ($\lambda_{ex}$: 485 nm; $\lambda_{em}$: 528 nm, transmission gain: optimal) using fluorescence plate reader (Tecan Infinite Pro F200). The resulting data were background-subtracted using the readings taken in the absence of ssDNA activator.

## DNA binding assay

MST was performed as described previously[4]. Briefly, dFnCas9-GFP and variant proteins were complexed with PAGE purified respective IVT sgRNAs (purified by 12% Urea-PAGE). The binding affinities of the Cas9 proteins and sgRNA RNP complexes were calculated using Monolith NT. 115 (NanoTemper Technologies GmbH, Munich, Germany). RNP complex (Protein:sgRNA molar ratio, 1:1) was reconstituted at 25⁰ C for 10 min in reaction buffer (20 mM HEPES, pH 7.5, 150 mM KCl, 1 mM DTT, 10 mM MgCl₂) HPLC purified 30 bp dsDNA (IDT) of different genomic loci with varying concentrations (ranging from

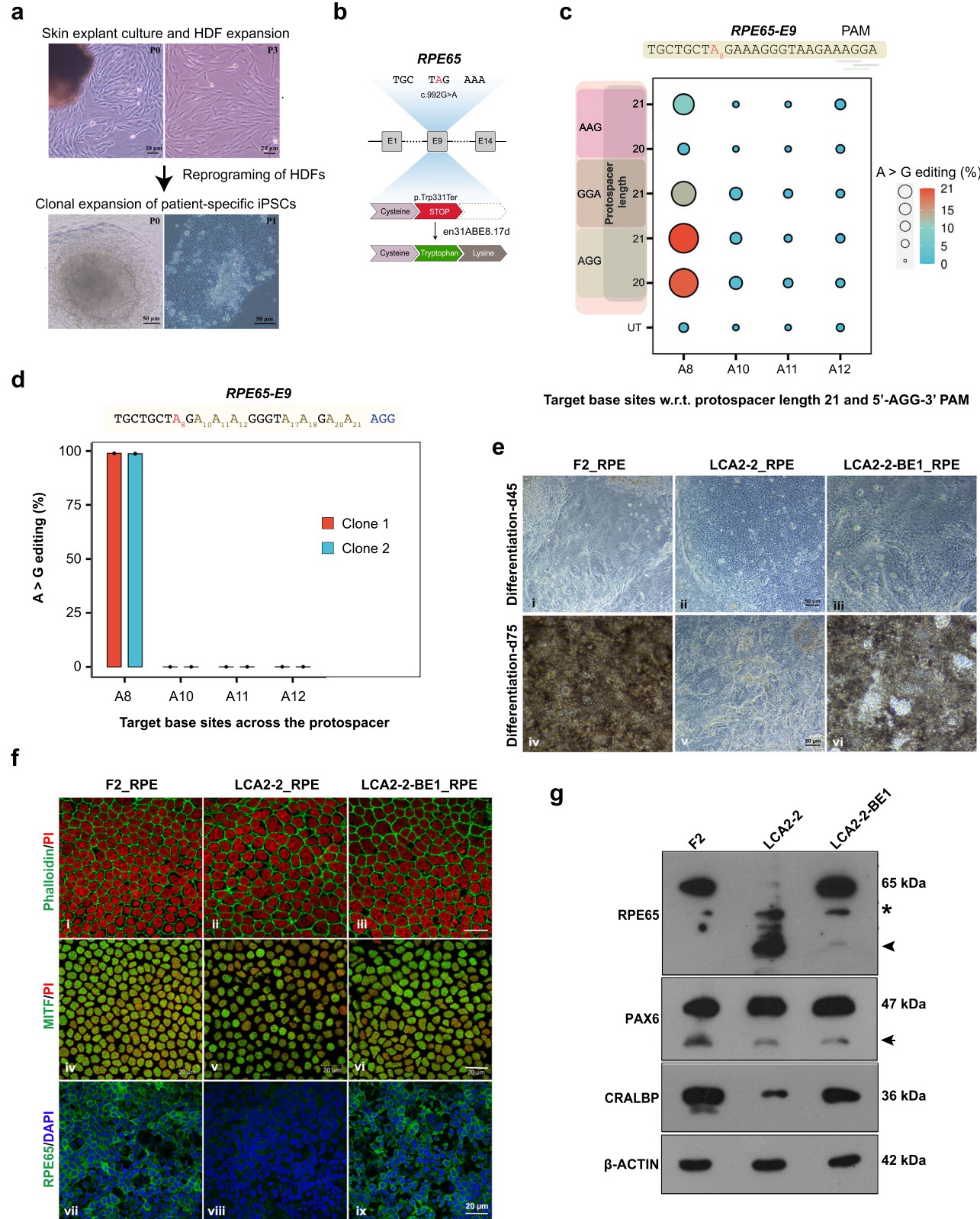

0.09 nM to 30 μM) were incubated with RNP complex at 37⁰ C temperature for 30 min in reaction buffer. The sample was loaded into NanoTemper standard treated capillaries and measurements were performed at 25 °C using 20% LED power and 40% MST power. Data analyses were done using NanoTemper analysis software and the data were plotted by OriginLab.

## Cell culture and transfections

HEK293T cells were grown in DMEM media supplemented with high glucose (Invitrogen), 2 mM GlutaMax, 10% FBS (Invitrogen), 1x antibiotic and antimycotic (Invitrogen) at 37 °C in 5% CO2. Human iPS cells (LVP-F2-3F) were derived and maintained as described earlier[74]. Briefly, the cells were cultured using Essential 8™ complete media kit (Gibco,

**Fig. 7 | Therapeutic base editing by en31-ABEmax8.17d in a leber congenital amaurosis 2 patient-specific iPSC line (LCA2−2). a** Representative image showing iPSCs reprogrammed from the skin explant-derived human dermal fibroblasts (HDF) of an LCA2 patient (*n* = 1, P = passage number). A total of 12 clones were picked at P0 and clonally expanded. A stable patient-specific iPSC line (LCA2−2) at passage 12 was used for editing studies. **b** Schematic showing patient-specific point mutation correction by en31-ABEmax8.17d, targeting a premature stop codon in exon 9 of *RPE65* gene. **c** Ballon plot showing the A to G editing events (%) as obtained from amplicon sequencing upon targeting *RPE65-E9* with en31-ABEmax8.17d constructs encoding sgRNAs containing either 20- or 21-nt spacer sequences. The targeted PAM sequences, 5′-AGG-3′, 5′-GGA-3′ and 5′-AAG-3′ are underlined. Mean of *n* = 3 independent biological replicates are shown. **d** Bar plot showing the efficiency of A to G editing (%) plotted on the Y-axis, as obtained from amplicon sequencing of two en31-ABEmax8.17d-treated clonal lines, LCA2−2-BE1 (in red, clone 1), LCA2−2-BE2 (in blue, clone 2). Error bars represent mean ± SEM of *n* = 3 independent sequencing replicates with individual values shown as dots.

**c**, **d** Target bases are shown on the x-axis, and counted w.r.t. g21 and 5′-AGG-3′ PAM. The target sequence is shown and the pathogenic-mutation (A8) is highlighted in red. **e** Representative phase contrast images showing the lower magnification view of the cellular morphology and levels of pigmentation in iPSC-derived RPE cultures (*n* = 3) at days 45 (upper panels) and at day 75 (lower panels) of differentiation. **f** Representative immunoassayed iPSC-RPE culture images (*n* = 2), showing their morphology and expression of RPE-specific markers such as, Phalloidin-FITC (green), PI (red) (i-iii); MITF (green), PI (red) (iv-vi); and RPE65 (green), DAPI (blue) (vii-ix). **g** Representative western blot images showing the expression of different RPE-specific proteins in iPSC-derived RPE cell lysates of healthy control (F2), patient-specific (LCA2−2) and mutation-corrected (LCA2−2-BE1) lines (*n* = 2). β-ACTIN served as loading control. Asterisk (*) marks a non-specific band. Arrowhead marks the truncated RPE65 protein band in LCA2−2-RPE lysate. Arrow marks a minor isoform-specific band of PAX6. **a**, **e**, **f** Scale bars are as indicated. Raw data are provided in Source Data file.

Cat No. A1517001), along with the addition of 1x Penicillin-Streptomycin antibiotics solution (Gibco, Cat No. 15140122) and cultured on Vitronectin coated (Gibco, Cat No. A14700) cell culture plates at 37 °C in 5% CO2. The human RPE cell line, ARPE19 (ATCC, Cat No. CRL2302) was cultured in DMEM/F-12 medium (Gibco, Cat No. 10565018) supplemented with 10% FBS (Gibco, Cat No. 26140079) and 1x Penicillin-Streptomycin antibiotics solution (Gibco, Cat No. 15140122) at 37 °C in 5% CO2.

Transfection of HEK293T and ARPE19 cells were performed using Lipofectamine 3000 Reagent (Invitrogen, Thermo Fisher Scientific Inc.), following the manufacturer's protocol. For hiPSC lipofection, about 5 ×10⁴ cells were seeded onto Vitronectin-coated 24-well plates and cultured under standard hiPSC culture conditions. The adhered cells at 40-50% confluency were then transfected using the Lipofectamine™ Stem Transfection Reagent, as per the manufacturer's protocol (Invitrogen, Thermo Fisher Scientific Inc.).

### T7 endonuclease I assay

At 48 h post-transfection, the cells were lysed with 250 µL of extraction buffer (100 mM Tris pH 8.0, 1% SDS, 5 mM EDTA, 200 µg/mL Proteinase K) and incubated at 56 °C for 2 h and the genomic DNA was precipitated with the addition of isopropanol. The DNA pellet was washed with 70% ethanol, air-dried, and dissolved in TE buffer, pH 8.0. The target region of human *PAX6* exon 6 was amplified by PCR using the screening primer sets and DreamTaq DNA polymerase (Thermo Fisher Scientific Inc.), as per the manufacturer's protocol. The PCR amplicons were gel purified (Qiagen) and about 1 µg of the genomic DNA was subjected to denaturation at 95 °C for 5 min and renaturation by slow cooling in a dry thermostat. The annealed DNA amplicons with heteroduplexes were incubated with 1 µL of T7 endonuclease I (New England Biolabs) at 37 °C for 1 h. The cleaved DNA products in the reaction mix were separated by 8% agarose gel electrophoresis. Densitometry analysis was done using BioRad Image Lab software. The NHEJ event was calculated using the following formula:

$$\% \text{NHEJ events} = 100 \times \left[ 1 - (1 - \text{fraction cleaved})^{1/2} \right]$$

where, fraction cleaved = (density of digested product) / (density of digested product + density of undigested product).

The cleaved fraction was normalized for the transfection efficiency (% GFP⁺ᵛᵉ cells).

### ChIP sequencing (ChIP-seq)

HEK293T cells on 10 cm dishes were transfected with 30 µg of plasmids. 48 h post-transfection GFP-positive cells were FACS sorted (BD FACSMelody Cell Sorter). ChIP was done by essentially following the earlier reported protocol with modifications as per requirements of our experiments[24]. Sorted cells were cross-linked with 1%

formaldehyde (Sigma) with gentle rotation at room temperature for 15 min followed by quenching by adding 125 mM glycine. Cells were rinsed twice chilled PBS. Cells were centrifuged at 1500 x g for 10 min at 4 °C and the cell pellet was snap-frozen in liquid nitrogen before storing at −80 °C. The cell pellet was resuspended in pre-chilled 1 mL lysis buffer 1 (50 mM HEPES-KOH pH 7.5, 140 mM NaCl, 1 mM EDTA, 10% glycerol, 0.5% NP-40, 0.25% Triton X-100, 1x Roche protease inhibitor cocktail), rotated for 15 min at 4 °C and centrifuged at 1500 x *g* for 10 min at 4 °C. The pellet was resuspended in 1 mL pre-chilled lysis buffer 2 (10 mM Tris-Cl pH 8, 200 mM NaCl, 1 mM EDTA, 1x Roche protease inhibitor cocktail) and treated similarly as previous. Now, the nuclear pellet was resuspended in 500 µL pre-chilled sonication buffer (20 mM Tris-Cl pH 8, 150 mM NaCl, 2 mM EDTA, 0.1% SDS, 1% Triton X-100, 1x Roche protease inhibitor cocktail) and sonicated for 10 min using Covaris S220 focused ultrasonicator (duty factor 20%, duty cycle 5, PIP 140, CPB 200, water temperature 4 °C). The lysates were centrifuged by placing it in DNA LoBind microfuge tubes (Eppendorf) at maximum speed for 15 min at 4 °C and the supernatant was collected. 25 µL of lysate was saved as input (5%). Precleared diluted lysates were incubated with 5 ug anti-HA ChIP grade antibody (abcam #9110) overnight at 4 °C. The antibody-protein complexes were incubated with 15 µL of protein G magnetic beads (Dynabeads, Life Technologies) for 2 h at 4 °C. Beads were repeatedly washed using three of the buffers by adding pre-chilled ChIP dilution buffer, high salt buffer and LiCl buffer. Washed beads were next washed two times by TE buffer. The chromatin was recovered from the beads by incubating with the ChIP elution buffer for 15 min at room temperature with rotation. The eluted chromatin was reverse crosslinked, digested with Proteinase K treatment and contaminating RNA was removed by RNase followed by purification of DNA using ethanol precipitation. Purified DNA was tested for fold enrichment at sgRNA target region before library preparation for massively parallel sequencing. Sequencing libraries were prepared using NEBNext® Ultra™ II DNA Library Prep Kit by essentially following the manufacturer's protocol and sequenced on HiSeq X platform at MedGenome Labs Pvt. Ltd. (Bangalore, India).

### Amplicon sequencing

HEK293T cells on six-well dishes were transfected with 2 µg of respective Cas9-containing sgRNAs. 48 h post-transfection, GFP-positive cells were FACS sorted (BD FACSMelody Cell Sorter) and gDNA was isolated (Lucigen QuickExtract Extraction solution). PCR primers were designed, flanking the predicted double-stranded break site and amplified with Phusion High-Fidelity DNA polymerase (Thermo Fisher Scientific). The 16 S Metagenomic sequencing library preparation protocol was adapted for library preparation. Briefly, the respective loci were amplified using forward and reverse primers along with overhang adapter sequences using Phusion High-Fidelity DNA polymerase (Thermo Fisher). AMPure XP beads (A63881, Beckman

Coulter) were used to separate out amplicons from free primers and primer dimers. Dual indexing was done using Nextera XT V2 index kit followed by a second round of bead-based purification. The libraries were quantified using a Qubit dsDNA HS Assay kit (Invitrogen, Q32853) and were also loaded on agarose gel for the qualitative check. Libraries were normalized, pooled and were loaded onto the Illumina MiniSeq platform for 2 × 150 bp sequencing.

## Digenome-seq

Genomic DNA (gDNA) from HEK293T cells was purified using QIAmp DNA mini kit (Qiagen Cat No. 51306). 5 µg gDNA was incubated with either 500 nM respective Cas9 RNPs (Cas9:sgRNA, 1:2) or a mock sample containing nuclease-free water in 1x rCutSmart Buffer (NEB) and 5% glycerol in 500 µL reaction volume at 37 °C for 10 h. gDNA was purified using QIAmp DNA mini kit (Qiagen Cat No. 51306) after quenching the reactions with RNAse A (Ambion, 10 mg/mL) and Proteinase K (Invitrogen, 20 mg/mL). The purified gDNA was used for quantitative PCR using LightCycler 480 SYBR Green I Master (Roche) to confirm the on-target cleavage efficiencies w.r.t the mock sample, following which the gDNA of both the Cas9 treated and mock samples were sheared using S220 Focused-ultrasonicator (Covaris) with a median size of the DNA fragments around 250 to 300 bp (Fill 10, Duty 10, PWP 140, CPB 200, 120 sec, water temperature 4 °C). DNA library was prepared by NEBNext® Ultra™ II DNA Library Prep Kit for Illumina® (NEB #E7645) using the fragmented DNA essentially following the manufacturer's protocol with some modifications. Libraries were indexed by NEBNext® Multiplex Oligos for Illumina® (NEB #E7600) following limited cycle PCR as per the manufacturer's protocol. Indexed libraries were cleaned up using AMPure XP beads (A63881, Beckman Coulter). The libraries were quantified either by Qubit dsDNA HS Assay kit (Invitrogen, Q32853) or NEBNext® Library Quant Kit for Illumina® (NEB E7630) and pooled. The qPCR cycling conditions on the instrument, LightCycler® 480 System (Roche) were as follows: Initial denaturation 95℃ for 1 min followed by 35 amplification cycles of 95℃ for 15 sec; 63℃ for 45 sec and melt curve. 2 × 150 bp sequencing was performed on Illumina NovaSeq 6000 platform at 30-40x depth at LifeCell Diagnostics (Chennai, India).

## HDR assay at *DCX* locus in HEK293T

HEK293T cells were cultured in DMEM with GlutaMAX supplement (ThermoFisher Scientific Cat. No. 10566016) with 10% FBS serum. 70%-80% confluent HEK293T cells were harvested from a 6 well plate using Trypsin-EDTA (0.05%) (ThermoFisher Scientific Cat. No.: 25300062) and pipetted to make a single-cell suspension. For each electroporation reaction, a total 15ug plasmid was mixed in Resuspension buffer R, in which linearized donor plasmid DNA and Cas9-gRNA vector were taken in a 1:2 ratio. 6 ×10^5 cells were resuspended in 100 µL of Resuspension Buffer R containing plasmids and electroporation was performed using Neon Transfection System 100 µL Kit (ThermoFisher Scientific Cat. No. MPK10096) with double pulses at 950 V, 30 milliseconds pulse width. The electroporated cells were transferred immediately to a 6 well plate containing 2 mL of pre-warmed culture medium and incubated at 37 °C and 5% CO2. After 24 h, cells were washed and re-incubated with a fresh culture medium. 72 h post-electroporation GFP-positive cells per sample were sorted using BD FACSMelody Cell Sorter (BD Biosciences-US) and gDNA was isolated from the sorted cells using Wizard Genomic DNA Purification Kit (Promega) for qPCR genotyping. qPCR reactions were performed using LightCycler 480 SYBR Green I Master (Roche) added to 50 ng DNA for each sample. The cycling conditions on the instrument were as follows: Initial denaturation 95℃ for 5 min followed by 40 amplification cycles of 95℃ for 10 sec; 63℃ for 30 sec; 72℃ for 30 sec and melt curve. Log$_2$ fold change values were calculated by the $-2^{-\Delta\Delta Ct}$ method[89] for each sample with respect to untransfected control. A non-targeting region in genomic DNA was used for normalization.

## ChIP Seq analysis

Raw sequencing reads were mapped to the human reference genome GRCh38 using bowtie2[90]. Peaks were called over input samples using MACS2[91] with default parameters. Finally, scrambled sample peaks were used to remove background and false positive peaks from the dSpCas9 and dFnCas9 test samples. These filtered peaks were searched for off-targets based on sgRNA sequence homology with a maximum of 6 mismatches. On-target peak coverage plots were generated by the fluff profiles command with 'remove duplicates' option[92]. Overlap between the dSpCas9 and dFnCas9 ChIP peaks were calculated using bedtools[93] and plotted as weighted Venn diagrams with the help of Intervene[94].

## PAM frequency analysis

PAM frequencies were calculated for more than 167 Cas systems (146 unique PAM sequences) from the human reference genome (GRCh38.p13) using in-house python script.

## Amplicon sequencing analysis

Sequencing reads from different replicates were down-sampled prior to indel or base editing analysis for each target to remove sequencing read depth bias across the samples. Raw amplicon sequencing reads were subjected to indel frequency estimation for nucleases and A to G nucleotide conversion for adenine base editors using CRISPResso2 v2.0.45[95] with parameters such as ignoring substitutions for indel analysis and keeping minimum overlap between the forward and reverse read to the 10 bp for both the cases. The base editing data were visualized as balloon plots using a customized R script and the ggpubr package where the area of the dots is proportional to the magnitude of editing (numerical values).

## Digenome-seq analysis

Sequencing reads were quality-checked with FastQC v0.11.8 (https://www.bioinformatics.babraham.ac.uk/projects/fastqc/) and MultiQC v1.14[96]. Poor-quality bases and adapter sequences were removed using cutadapt v2.8[97] for paired-end reads. Trimmed reads with at least 45 bp length were retained. Processed paired-end reads were aligned to GRCh38.p14 with Bowtie2 v2.2.5[90] with default options and position-sorted with samtools v1.18[98]. Aligned reads to only canonical chromosomes were used for all downstream analysis. A command line version of the Digenome-seq web tool[99] was used for analysis with default options except, for SpCas9 and SpRY the overhang value of 0 and for enFnCas9 enzymes 3 was used. Cut-off for Digenome-seq score was manually checked from depth of aligned reads for all the cas9 enzymes. Hits from Digenome-seq were further filtered with an in-house Python script to exclude false positives. The analysed data was plotted with circos v0.69-8[100] and the intersection of the captured sites across Cas9s was plotted with UpsetR v.1.4.0[101].

## Analysis of targetable pathogenic SNPs for Adenine base editors (ABEs)

A variant summary file was downloaded from ClinVar FTP sites in March 2023 to check for targetable G > A variations in ClinVar. Several pre-processing steps were performed to filter the data before evaluating the editable adenine alternate base in ClinVar. Initially, only mutations reported in the 'GRCh38' assembly were kept, and then only single nucleotide mutations with clinical significance for pathogenicity were chosen. Only mutations with the alternate allele 'A' and the reference allele 'G' were then allowed to proceed. The positions of mutations were used to map them onto the human reference genome after pre-processing (GRCh38.p13). Furthermore, the PAM sequences (which differed between base editors) were located near the mutations, as was the base editing window, which differed between base editors. For the calculation of the targetable adenine mutations, pathogenic (G > A) mutations that met the above criteria were

considered. Finally, the targetability of various base editors was plotted using Bioconductor's circlize package.

## Western blotting

Cells were lysed using the RIPA lysis buffer (Pierce™, Thermo Fisher Scientific Inc.) in the presence of 1x Protease Inhibitor Cocktail (PIC), (Roche, Sigma-Aldrich), to prepare the whole cell lysates. The concentration of the lysates of each sample was estimated using the Pierce™ BCA Protein Assay Kit (Thermo Fisher Scientific Inc.). 25 µg of lysate for each sample was subjected to SDS-PAGE followed by standard blot transfer and washing methods, before imaging the blots using the Syngene Gel documentation instrument. Antibodies used in the experiments are given in the respective figure legends and antibody details are mentioned in the Supplementary Data 4.

## Immunostaining and confocal imaging

The ARPE19 cells grown on glass coverslips were washed with phosphate-buffered saline (1x PBS) 48 h after transfection, fixed with 3.5% formaldehyde in 1x PBS for 10 min, followed by three washes with 1x PBS for 5 min each. The cells were then permeabilized with 0.5% Triton X-100 in 1x PBS for 10 min, followed by three 1x PBS washes, and then blocked with 10% FBS in 1x PBS (blocking buffer) for 1 h. The cells were then sequentially incubated with specific primary antibodies diluted in blocking buffer for 1 h and washed three times with 1x PBS, followed by incubation with species-specific secondary antibodies conjugated to different fluorophores for 45 min. The samples were then washed with 1x PBS, counterstained with DAPI, and mounted onto a glass slide using the Vectashield mountant (Vector Laboratories). The samples were imaged and analysed using the Zeiss LSM 880 confocal laser scanning microscope and Zeiss Zen Blue software and the images were assembled as collage using the Adobe Photoshop CS6.

## Genetic screening and identification of mutation in LCA2 patients

As a part of an ongoing genetic screen of a large case series, the proband (LCA27) was clinically diagnosed as a candidate for Leber congenital amaurosis, type 2 (LCA2), based on the clinical characteristics described earlier[102]. The genomic DNA was extracted from the blood sample of all patients and were genotyped for a select set of LCA candidate genes, by targeted amplification of all exons, along with the flanking intronic regions by PCR, followed by Sanger sequencing. The proband, was found to carry a pathogenic, nonsense mutation within the exon 9 of the human *RPE65* gene, c.992 G > A, p.Trp331Ter (TGG > TAG), which resulted in a stop codon, premature translational termination, which led to a C-terminally truncated and non-functional protein[71]. This patient volunteer, a 24-year-old male of Asian-Indian origin, was counseled with duly executed informed consent and recruited for skin biopsy collection, to generate human dermal fibroblast cultures, which were further reprogrammed to generate the patient-specific iPSC line, LVPEIi005-A (LVIP02-LCA2-2).

## HDF culture, reprogramming and generation of hiPSC lines

A full-thickness skin biopsy of about 2 × 2 mm was taken from the retro-auricular surface of the patient volunteer, with the informed consent. The tissue was cut into small pieces and digested with Collagenase (1 mg/mL) for 3 h, diluted with DPBS and then centrifuged at 200 x g for 3 min. The pellet containing the released cells, along with the partially digested tissues were suspended in 5 mL of HDF culture medium (Gibco™ Medium 106, with the addition of Low Serum Growth Supplement- LSGS, Thermo Fisher Scientific Inc.) and cultured in T25 flasks at 37 °C with 5% CO$_2$ for 3-5 days or until the culture reaches 80-90% confluence. The human dermal fibroblast (HDFs) cultures were further passaged using 1x TrypLE at 1:3 split ratio. For HDF reprogramming, a cocktail of three episomal plasmids encoding the human OCT3/4, SOX2, KLF4, L-MYC and LIN28 (Addgene Plasmid # 27077, 27078, 27080)[103] were nucleofected into passage 3 HDFs, using the P2 nucleofection kit and EO-114 program of the 4D-Nucleofector, X Unit System, as per the manufacturer's instructions (Lonza, Basel, Switzerland). The cells were then plated onto vitronectin-coated cell culture dishes and cultured in HDF medium for the first 2 days. On the third day, the cultures were shifted to hiPSC growth medium (Gibco™ Essential 8™ medium, Thermo Fisher Scientific Inc.) and maintained till day 30, with regular changes of medium on alternate days. Well reprogrammed hiPSC colonies with distinct margins emerged at around D20-D25. Individual colonies were manually picked by gentle nudging using a P10 tips and are mildly triturated to form 5-10 cell clusters and further cultured on vitronectin-coated 12-well plates. Well expanding hiPSC clones were further passaged using ReLeSR (STEMCELL Technologies Inc.) at 1:6 split ratio till passage 10, for further expansion, molecular characterizations to confirm stemness, pluripotency, genetic identity, genomic stability and for the loss of episomal plasmids. The cells of stable patient-specific iPSC lines at passage 12 were used in base editing experiments. The stable patient-specific iPSC line, LVPEIi005-A (LVIP02-LCA2-2) was characterized for the genetic identity (patient-specific mutation, STR profiling), stemness marker expression (by RT-PCR, Immunocytochemistry) and pluripotency (by EB formation and three lineage marker expression by RT-PCR). The iPSC lines registered at hPSCreg: healthy control iPSC, LVPEIi001-B (LVIP02-NC-F2-1), patient-specific mutant iPSC, LVPEIi005-A (LVIP02-LCA2-2) and a mutation-corrected, patient-specific iPSC clone, LVPEIi005-A-1 (LVIP02-LCA2-2-BE1)

## Base editing and mutation correction in patient-specific iPSCs

The patient-specific iPSC culture was harvested using ReLeSR (STEMCELL Technologies Inc.) to prepare single cell suspensions and the cell count was taken using a haemocytometer. About 1 million cells were prepared for the nucleofection of en31-ABEmax8.17d plasmids with cloned mutation-specific gRNAs containing either 20- or 21-nt spacers, using the P3 nucleofection kit and CA-137 program of the 4D-Nucleofector, X Unit system, as per the manufacturer's instructions (Lonza, Basel, Switzerland). After nucleofection, the cells were suspended in hiPSC growth medium (Gibco™ Essential 8™ medium, Thermo Fisher Scientific Inc.), with the addition of Gibco™ RevitaCell™ supplement to support single cell culture and were plated onto two wells of a vitronectin-coated 6-well plate. After 18-24 h, the cultures were shifted to the complete Essential 8 medium, with regular changes of medium on alternate days and were maintained in a controlled condition incubator at 37 °C, with 5% CO$_2$ supply. At 72 h post nucleofection or when the cultures reach 70-80% confluence, one well of the edited cell pool was harvested using ReLeSR and cryopreserved using Gibco™ Synth-a-Freeze™ cryopreservation medium. The second well was used for DNA isolation to evaluate edit efficiencies, either by T7 endonuclease I assay or by deep sequencing of PCR amplicons of the targeted genomic region. Once the presence of edit(s) were confirmed, a vial of cryopreserved cell pool was revived and plated at clonal densities ( < 1000 cells per 100 mm dish) on vitronectin-coated plates and cultured in Essential 8 medium with the addition of RevitaCell supplement during the initial 18-24 h of single cell passaging. Well grown single cell clones with clear margins (n = 24), were manually picked and individually plated onto vitronectin-coated 12-well plates and further cultured for clonal expansion. Subsequent clonal passage was done using ReLeSR for harvesting the cells and replica plates were maintained. The harvested cells were used for further expansion, cryopreservation, and DNA isolation for genomic edit analysis. Two of the clones that were genotyped and confirmed to carry the desired edits are sequentially expanded to generate mutation corrected, patient-specific clonal lines, LVIP02-LCA2-2-BE1, BE2 at passage 20.

## Differentiation of hiPSCs into mature RPE cells

A healthy control iPSC, LVPEIi001-B (LVIP02-NC-F2-1), patient-specific mutant iPSC, LVPEIi005-A (LVIP02-LCA2-2) and a mutation-corrected, patient-specific iPSC clone, LVPEIi005-A-1 (LVIP02-LCA2-2-BE1) were differentiated into early eye fields and mature retinal lineages, as described earlier[74,75]. Briefly, the near confluent and adherent cultures of hiPSCs grown in E8 medium on Matrigel-coated dishes were directly induced to differentiate with the addition of Differentiation Induction Medium ($n = 3$) [DIM: Basal differentiation medium (DMEM-F12, 10% Knockout Serum Replacement (KOSR), 1x Non-Essential Amino Acid (NEAA), 2 mM GlutaMax, 100 U/mL Penicillin-Streptomycin, 200 μM L-Ascorbic acid), containing 1% N2 supplement, 1 ng/mL bFGF and 1-10 ng/mL Noggin], for the first 3 days, with gradual withdrawal of bFGF and increasing concentrations of Noggin. On day 4 (d4), the cultures were shifted to the Retinal Differentiation Medium [RDM: Basal differentiation medium containing 2% B27 supplement] and were continuously maintained for 3-4 weeks at 37 °C in a 5 % $CO_2$ incubator, with daily changes of medium. At around 21-28 days, self-organized, distinct eye-field primordial clusters (EFPs) were observed, with a central island of circular 3D neuro-retinal structures surrounded by contiguous outgrowths of neuro epithelium and ocular surface epithelium. Using a flame pulled glass Pasteur pipette with a hooked tip, the intact neuro-retinal cups were scooped out and further cultured in RDM and maintained as suspension cultures in non-adherent dishes for up to d60 to generate neuro-retinal organoids. The organoid cultures were further maintained in RDM containing 100 μM Taurine for up to 90-120 days, to enable further maturation and photoreceptor cell differentiation. After the harvest of neuro-retinal cups at d21-28, the differentiation cultures were continuously maintained in RDM for up to d45, till the mildly pigmented and proliferating retinal pigmented epithelium (RPE) patches emerged surrounding the central NR islands. These pigmented RPE patches were then manually picked for enrichment, as described above and cultured on Matrigel or Laminin 521 or Collagen I-coated culture surfaces in RPE medium [RPE-M: Basal differentiation medium containing 2% B27 supplement, 10 ng/mL Activin A, 10 mM Nicotinamide and 10 μM Y-27632] till d60. The RPE cultures were further maintained in Retinal Pigmented Epithelial Maturation Medium [RPE-MM: Basal differentiation medium containing 1% N2 supplement, 1x THT (2 mM Taurine, 55 nM Hydrocortisone, 20 pM Triiodothyronine) and 1 μM PD0325901], with media changes on alternate days till d75, to establish the enriched RPE cell cultures, for further experimentation or for cryopreservation. Alternately, the cultures were continuously maintained in RPE-MM for extended periods of up to 90-120 days, to establish fully mature and well polarized, pigmented RPE cell cultures.

## Karyotyping

Growing iPSC cultures at about 70-80% confluence was treated with sterile colcemid (0.1 μg/mL; Sigma-Aldrich) for 2-3 h to induce metaphase arrest. The cells were then harvested using 0.25% trypsin to prepare single-cell suspensions and further treated with a hypotonic solution (0.075 M KCl), fixed using ice-cold methanol and glacial acetic acid (3:1) and then dropped onto clean glass slides (Thermo Fisher Scientific Inc.) and air dried. After a brief trypsin treatment, the chromosomes were G-banded by Giemsa staining. Clear metaphase spreads ($n = 20$) were imaged and analysed using the CytoVision automated image analysis system (Applied Imaging).

## Statistics and reproducibility

Statistical details are mentioned in the figure legends. For all analyses, $p$-values ≤ 0.05 were considered statistically significant. $p$-values are represented for * ≤0.05, ** ≤0.01, *** ≤0.001.

## Reporting summary

Further information on research design is available in the Nature Portfolio Reporting Summary linked to this article.

## Data availability

Deep sequencing data from ChIP-seq, amplicon sequencing, and Digenome-seq experiments are deposited as a BioProject under Project ID PRJNA766155. The analysed genomic coordinates from Digenome-seq are available in the Source Data file. Primers and antibody details used in this study are provided in Supplementary Data 3 and 4, respectively. Data from Fig. 1c, Supplementary Fig. 5b-e are available at doi.org/10.6084/m9.figshare.25827652. Data from Supplementary Fig. 10d and e are available at doi.org/10.6084/m9.figshare.25827808. Source data are provided with this paper.

## Code availability

Codes used for Fig. 2b and Supplementary Fig. 9 are available at doi.org/10.6084/m9.figshare.25866661. Other codes used in this study are available upon request.

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

## Acknowledgements

We thank all members of Chakraborty, Maiti, Nureki, Mariappan and Nishimasu labs for helpful discussions and valuable insights pertaining to this work. This study was funded by CSIR Sickle Cell Anemia Mission (HCP0008 and HCP0023) and Lady Tata Young Investigator award (GAP0198) to D.C., Science and Engineering Research Board (SB/SO/HS/177/2013) and Department of Biotechnology (BT/PR32404/MED/30/2136/2019) to I.M. and Senior Research Fellowships from ICMR (S.A., Su.M.), UGC (T.A.) and CSIR (V.K.P.), Government of India. S.A. acknowledges the support from Stellar Science Foundation (SS-F), Tokyo, Japan, during the revision of this study.

## Author contributions

S.A., S.M., and D.C. conceived the project and designed the experimental pipeline. S.A. designed and performed the protein engineering experiments with inputs from S.H., H.N., and O.N. S.A. optimized and performed ChIP-seq assay, cellular editing assay, Digenome-seq assay, and all the sequencing experiments. A.H.A. designed and implemented bioinformatics tools for the design, validation, and execution of enFnCas9-based diagnostics and editing experiments with inputs from S.A. and D.C. P.K.D. designed and implemented the bioinformatics pipeline for analysing Digenome-seq data with inputs from S.A. and D.C. M.Ai. contributed to Digenome-seq experiments and a part of cellular editing experiments using enFnCas9 variants. R.R. performed HDR experiments with enFnCas9 variants. S.S. performed a comparison of

single mismatch specificity with Cas12/14 proteins, cloning of guides in base editing constructs, and assisted in sequencing experiments. M.K., R.P., and S.G. contributed to validating the FELUDA/RAY platform with enFnCas9 proteins. A.R. purified the SpRY and Superfi-Cas9 protein and performed in vitro assays. A.G. and C.A. performed in vitro assays. D.P. contributed to a part of cellular editing experiments using enFnCas9 variants. S.A., Su. M, Sa. M, V.K.P, and I.M. performed and validated enFnCas9 activity experiments in HEK293T, ARPE-19, iPSC lines, and their characterizations. T.A. contributed to the RPE analysis. S.J. and I.M. contributed to the LCA patient resource. S.A. and D.C. drafted the manuscript with inputs from all other authors. Correspondence should be addressed to S.A. and D.C.

## Competing interests

S.A., S.M., and D.C. are listed as co-inventors for US Patent titled, "Kinetically enhanced engineered FnCas9 and its uses thereof" related to the engineered FnCas9 variants described in this study. The patent applicant is the Council of Scientific and Industrial Research, New Delhi, India, and the application number is 18/049,291. The patent has been granted. The other authors declare no competing interests.
