## [Peer Review File · Nature Communications]

Reviewers' Comments:

Reviewer #1:

Remarks to the Author:

In this revised manuscript entitled "Combination of engineered FnCas9 and extended gRNAs for PAM-flexible, robust and nucleobase specific editing and diagnostics", the authors have addressed most of the technical concerns raised by Reviewers #1 and #2. However, the overall readability of the revised manuscript requires further improvement, including the formatting and logic. Some specific suggestions are listed below for authors' consideration.

Specific concern:

1. Not clear why including so many details about the experimental results in the introduction section.
2. Please reorganize to streamline the structure of the manuscript with concise and clear logic. For example, shorten 10 main figures to no more than 6-7. Some figures and sup figures can be combined.
3. Figure 1C was not cited in the main text of the revised manuscript. Sup Figure 4D was described before sup Figure 4A-4C.
4. There are three and half pages of details for Figure 1 together with four sup Figures are relevant to Figure 1. Please consider shorten to remove redundant details.
5. Many mis-labelling and mistakes in this revised manuscript, including:
 - a) in Lines 150-151, it reads as " between dC(-2) and dA(-1) in the target DNA strand and between dG(2) and dT(1) in the non-target strand respectively (Supplementary Figure 3D) ", however the relevant sup Figure 3D only shows dA(-1) and dT*, but no dC(-2), dG(2) or dT(1).
 - b) Please keep consistency in the format for 5'-NGA-3' and 5'-NGA3'throughout the manuscript.
 - c) In line 208, what is the meaning of "48 = 65,536 combinations in total"?
 - d) Seems like that are unwanted line breaks in line 309, line 393 and line 515.
 - e) Please keep consistency for the writing format when describing multiple figures, for example: (Figure 5D, E) in line 383 and (Figure 6A-B, Supplementary Figure 9A-B) in line 387.

Reviewer #3:

Remarks to the Author:

In this revision, Acharya, et al., have improved this manuscript with additional experiments proving the low off-targeting and the flexible editing of enFnCas9. While the paper is stronger with this new data, I am still not convinced that enFnCas9 will be adopted as a useful genome editing tool, as the discrimination between previously-engineered variants has not been thoroughly conducted. Still, I appreciate the authors conducting Digenome-Seq and demonstrating the advantageous off-targeting profile of their enzyme.

In the response to reviewer comments (both mine and the other reviewers), rather than perform direct comparisons to SpCas9, SpRY, or other orthologs (eNme2Cas9, Sc++, etc.), the authors compared the on- and off-targeting data in those papers directly to their data. At least some additional comparisons (PAM profile are necessary to help readers place enFnCas9 as a useful tool. However, the authors argue that additional comparisons are not necessary.

Outside of this, I think the manuscript conclusions are sufficiently well-justified.

Reviewer #1 (Remarks to the Author):

In this revised manuscript entitled "Combination of engineered FnCas9 and extended gRNAs for PAM-flexible, robust and nucleobase specific editing and diagnostics", the authors have addressed most of the technical concerns raised by Reviewers #1 and #2. However, the overall readability of the revised manuscript requires further improvement, including the formatting and logic. Some specific suggestions are listed below for authors' consideration.

Response:

We thank Reviewer 1 for his/her critical reading of our manuscript and being satisfied with our addressal of most of the technical questions. We take the point of readability very seriously and have accordingly made changes in the manuscript to improve formatting and logic.

Specific concern:

1. Not clear why including so many details about the experimental results in the introduction section.

Response:

We thank the Reviewer 1 for this pertinent point. To address this, we have shifted most of the experimental details to the result section in the current manuscript.

2. Please reorganize to streamline the structure of the manuscript with concise and clear logic. For example, shorten 10 main figures to no more than 6-7. Some figures and sup figures can be combined.

Response:

We thank the Reviewer 1 for this point. To address this, we have reformatted the structure of the manuscript following a clear logic of protein engineering, validation and therapeutics. Also we have made sure there are only 7 figures and several figures and supplementary figures have been compiled.

3. Figure 1C was not cited in the main text of the revised manuscript. Sup Figure 4D was described before sup Figure 4A-4C.

We thank the Reviewer 1 for this point and apologize for this inadvertent error. Supplementary Figure 4D is now 4A and has been described in the manuscript.

4. There are three and half pages of details for Figure 1 together with four sup Figures are relevant to Figure 1. Please consider shorten to remove redundant details.

We thank the Reviewer 1 for this suggestion. However to the best of our efforts we were not able shorten details in Figure 1 since it forms the crux of the protein

engineering work. We hope with the shortening of additional figures, the brevity aspect is now addressed.

5. Many mis-labelling and mistakes in this revised manuscript, including:
a) in Lines 150-151, it reads as “ between dC(-2) and dA(-1) in the target DNA strand and between dG(2) and dT(1) in the non-target strand respectively (Supplementary Figure 3D) ” , however the relevant sup Figure 3D only shows dA(-1) and dT*, but no dC(-2), dG(2) or dT(1).

We thank the Reviewer 1 for pointing these out. We have now corrected all the points in the submitted manuscript.

b) Please keep consistency in the format for 5'-NGA-3' and 5'-NGA3'throughout the manuscript.

We thank the Reviewer 1 for pointing this. We have addressed the consistency issue in the revised manuscript.

c) In line 208, what is the meaning of “48 = 65,536 combinations in total”?

We thank the Reviewer 1 for pointing this typographical error. We have addressed the same in the revised manuscript.

d) Seems like that are unwanted line breaks in line 309, line 393 and line 515.

We thank the Reviewer 1 for pointing this copyediting error and have corrected the same.

e) Please keep consistency for the writing format when describing multiple figures, for example: (Figure 5D, E) in line 383 and (Figure 6A-B, Supplementary Figure 9A-B) in line 387.

We thank the Reviewer 1 for pointing these errors and have corrected the same in the revised manuscript.

Reviewer #3 (Remarks to the Author):

In this revision, Acharya, et al., have improved this manuscript with additional experiments proving the low off-targeting and the flexible editing of enFnCas9. While the paper is stronger with this new data, I am still not convinced that enFnCas9 will be adopted as a useful genome editing tool, as the discrimination between previously-engineered variants has not been thoroughly conducted. Still, I appreciate the authors conducting Digenome-Seq and demonstrating the advantageous off-targeting profile of their enzyme.

In the response to reviewer comments (both mine and the other reviewers), rather than perform direct comparisons to SpCas9, SpRY, or other orthologs (eNme2Cas9, Sc++, etc.), the authors compared the on- and off-targeting data in those papers directly to their data.

Response:

We thank Reviewer 3 for his/her critical reading of the manuscript and for appreciating aspects of the advantageous off-targeting profile of our enzyme. We however respectfully disagree with the speculation that enFnCas9 will be adopted as a useful genome editing tool in the light of the LCA editing in patient iPSCs that we have provided and additional clinical studies that are currently undergoing. We are cautiously optimistic that the outcomes of these studies will advance the field of therapeutic translation of precise genome editing proteins like enFnCas9.

We would like to point out that the Reviewer might have missed that we have indeed done a direct comparison with SpCas9, SpRY in Figure 5, Supplementary Figure 7 and show how enFnCas9 variants outperform the other orthologs in their specificity and activity. We are happy to have completed these experiments to further strengthen our claims.

At least some additional comparisons (PAM profile are necessary to help readers place enFnCas9 as a useful tool. However, the authors argue that additional comparisons are not necessary.

Outside of this, I think the manuscript conclusions are sufficiently well-justified.

We thank the Reviewer for his/her suggestions. We have provided in vitro PAM profile in the manuscript already in addition to cellular validation of activity at alternate PAMs for both nuclease and base editors. In our opinion we have addressed this point significantly through the additional experiments. We appreciate the Reviewer for thinking that the manuscript conclusions are sufficiently well-justified.